# Widespread discrepancy in *Nnt* genotypes and genetic backgrounds complicates granzyme A and other knockout mouse studies

Daniel J Rawle[1†], Thuy T Le[1†], Troy Dumenil[1], Cameron Bishop[1], Kexin Yan[1], Eri Nakayama[1,2], Phillip I Bird[3], Andreas Suhrbier[1,4]*

[1]QIMR Berghofer Medical Research Institute, Brisbane, Australia; [2]Department of Virology I, National Institute of Infectious Diseases, Tokyo, Japan; [3]Department of Biochemistry and Molecular Biology, Biomedicine Discovery Institute, Monash University, Melbourne, Australia; [4]Australian Infectious Disease Research Centre, GVN Center of Excellence, Brisbane, Australia

**\*For correspondence:**
Andreas.Suhrbier@
qimrberghofer.edu.au

[†]These authors contributed equally to this work

**Competing interest:** The authors declare that no competing interests exist.

**Abstract** Granzyme A (GZMA) is a serine protease secreted by cytotoxic lymphocytes, with *Gzma*[-/-] mouse studies having informed our understanding of GZMA's physiological function. We show herein that *Gzma*[-/-] mice have a mixed C57BL/6J and C57BL/6N genetic background and retain the full-length nicotinamide nucleotide transhydrogenase (*Nnt*) gene, whereas *Nnt* is truncated in C57BL/6J mice. Chikungunya viral arthritis was substantially ameliorated in *Gzma*[-/-] mice; however, the presence of *Nnt* and the C57BL/6N background, rather than loss of GZMA expression, was responsible for this phenotype. A new CRISPR active site mutant C57BL/6J *Gzma*[S211A] mouse provided the first insights into GZMA's bioactivity free of background issues, with circulating proteolytically active GZMA promoting immune-stimulating and pro-inflammatory signatures. Remarkably, k-mer mining of the Sequence Read Archive illustrated that ≈27% of Run Accessions and ≈38% of BioProjects listing C57BL/6J as the mouse strain had *Nnt* sequencing reads inconsistent with a C57BL/6J genetic background. *Nnt* and C57BL/6N background issues have clearly complicated our understanding of GZMA and may similarly have influenced studies across a broad range of fields.

## Editor's evaluation

This paper is of great interest as a serendipitous discovery that, in the course of investigating the physiological role of granzyme A, has revealed the significance of the *Nnt* gene mutation in the inflammatory responses in mouse models. For many researchers in the fields of medicine and biology using C57BL/6 mice, the data obtained in this study will provide a useful opportunity to revisit previous findings and to gain new insights.

## Introduction

Granzyme A (GZMA) is a granule trypsin-like serine protease (trypase) secreted by cytotoxic lymphocytes such as NK cells (*Fehniger et al., 2007*; *Wu et al., 2019*), NKT cells (*Gordy et al., 2011*), and CD8+ cytotoxic T lymphocytes (*Suhrbier et al., 1991*). The traditional view has been that GZMA is a cytotoxic mediator that is secreted into the immunological synapse, entering the target cell via perforin pores, whereupon certain cytoplasmic proteins are cleaved, resulting in the initiation of cell death pathway(s) (*Liesche et al., 2018*; *Martinvalet et al., 2008*; *Wu et al., 2019*; *Zhou et al., 2020*).

A key tool in the quest to understand the physiological role of GZMA has been the use of *Gzma*⁻/⁻ mice (*Ebnet et al., 1995*). For instance, control of viral infections can be compromised in *Gzma*⁻/⁻ mice (*Loh et al., 2004*; *Müllbacher et al., 1996*; *Pereira et al., 2000*; *Riera et al., 2000*), with cytotoxic lympho-cytes from these mice reported to be less able to kill target cells (*Pardo et al., 2002*; *Pardo et al., 2004*; *Shresta et al., 1999*; *Susanto et al., 2013*). Like granzyme B (GzmB), GZMA has thus been classified as a cytotoxic granzyme (*Golstein and Griffiths, 2018*; *Mpande et al., 2018*; *Muraro et al., 2017*; *Zhou et al., 2020*), although in several studies a role for GZMA in mediating cellular cytotoxicity was not observed (*Ebnet et al., 1995*; *Joeckel and Bird, 2014*; *Regner et al., 2009*; *Regner et al., 2011*; *Smyth et al., 2003*). In a range of settings, GZMA has also been associated with the promo-tion of inflammation, providing an additional or alternative view of its physiological role, although consensus on mechanisms has remained elusive (*Metkar et al., 2008*; *Park et al., 2020*; *Santiago et al., 2020*; *Santiago et al., 2017*; *Schanoski et al., 2019*; *Shimizu et al., 2019*; *van Daalen et al., 2020*; *Wensink et al., 2015*; *Wilson et al., 2017*), with a number of potential intracellular and extra-cellular targets for GZMA reported. These include pro-IL-1β (*Hildebrand et al., 2014*), SET complex proteins (*Mandrup-Poulsen, 2017*; *Mollah et al., 2017*), gasdermin B (*Zhou et al., 2020*), mitochon-drial complex I protein NDUFS3 (*Martinvalet et al., 2008*), protease activated receptors (*Hansen et al., 2005*; *Sower et al., 1996*; *Suidan et al., 1994*; *Suidan et al., 1996*), TLR2/4 (*van Eck et al., 2017*), and TLR9 (*Shimizu et al., 2019*). *Gzma*⁻/⁻ mice have also been used to show a role for GZMA in inter alia diabetes (*Mollah et al., 2017*), cancer (*Santiago et al., 2020*), bacterial infections (*García-Laorden et al., 2016*; *García-Laorden et al., 2017*; *van den Boogaard et al., 2016*), and arthritis (*Santiago et al., 2017*). GZMA's bioactivity has generally (*Plasman et al., 2014*; *Schanoski et al., 2019*; *Zhou et al., 2020*), but not always (*Shimizu et al., 2019*; *van Eck et al., 2017*), been associated with GZMA's protease activity, with circulating GZMA in humans shown to be proteolytically active (*Spaeny-Dekking et al., 1998*).

After infection with the arthritogenic alphavirus, chikungunya virus (CHIKV) (*Suhrbier, 2019*), infected *Gzma*⁻/⁻ mice showed a substantially reduced overt arthritic foot swelling when compared to infected C57BL/6J (6J) mice (*Wilson et al., 2017*). However, we show herein that active site mutant *Gzma*^S211A mice generated by CRISPR on a 6J background showed no significant differences in CHIKV arthritic foot swelling when compared with 6J mice. The apparent contradiction was resolved when it emerged that *Gzma*⁻/⁻ mice had a mixed 6J and C57BL/6N (6N) background. *Gzma*⁻/⁻ mice retained expression of the full-length nicotinamide nucleotide transhydrogenase (*Nnt*) gene, whereas 6J mice have a truncated *Nnt* with a 5 exon deletion (*Fontaine and Davis, 2016*). The presence of full-length *Nnt* and the mixed 6N/6J genetic background (rather than absence of GZMA expression) emerged to be responsible for amelioration of CHIKV arthritic foot swelling in *Gzma*⁻/⁻ mice. As much of our under-standing of the physiological role of GZMA comes from studies in *Gzma*⁻/⁻ mice, we used the *Gzma*^S211A mice to gain new insights into GZMA function that were not compromised by genetic background.

The enzyme, nicotinamide nucleotide transhydrogenase (NNT), also known as proton-translocating NAD(P)+ transhydrogenase (EC 7.1.1.1), is located in the inner mitochondrial membrane and catalyzes the conversion of NADH plus $NADP^+$ to $NAD^+$ plus NADPH, while $H^+$ is pumped from the inter-membrane space into the mitochondrial matrix (*Rydström, 2006*). NNT thereby sustains mitochondrial antioxidant capacity through generation of NADPH (*Ward et al., 2020*), with the loss of active NNT in 6J mice associated with reduced ability to detoxify reactive oxygen species (ROS) via the glutathione and thioredoxin pathways (*McCambridge et al., 2019*; *Meimaridou et al., 2018*; *Ronchi et al., 2013*; *Rydström, 2006*). As redox regulation is involved in many cellular processes (*Gambhir et al., 2019*; *Lingappan, 2018*; *Sun et al., 2020*) and genetic backgrounds are known to affect phenotypes (*Leskov et al., 2017*; *Morales-Hernandez et al., 2018*; *Rao et al., 2020*; *Salerno et al., 2019*; *Vozenilek et al., 2018*; *Williams et al., 2021*; *Wolf et al., 2016*), we sought to determine how many other studies in GMO mice might have been affected by 6J vs. 6N background differences using the *Nnt* gene as a genetic marker. k-mer mining of RNA-Seq datasets deposited in the NCBI Sequence Read Archive (SRA) revealed that ≈27% of Run Accessions and ≈38% of BioProjects listing the mouse strain as 'C57BL/6J' had *Nnt* reads inconsistent with a 6J background. Although reported as underappreciated in the metabo-lism literature (*Fontaine and Davis, 2016*), potential problems associated with differences in *Nnt* and/or other background genes clearly extends well beyond this field and are not restricted to *Gzma*⁻/⁻ mice.

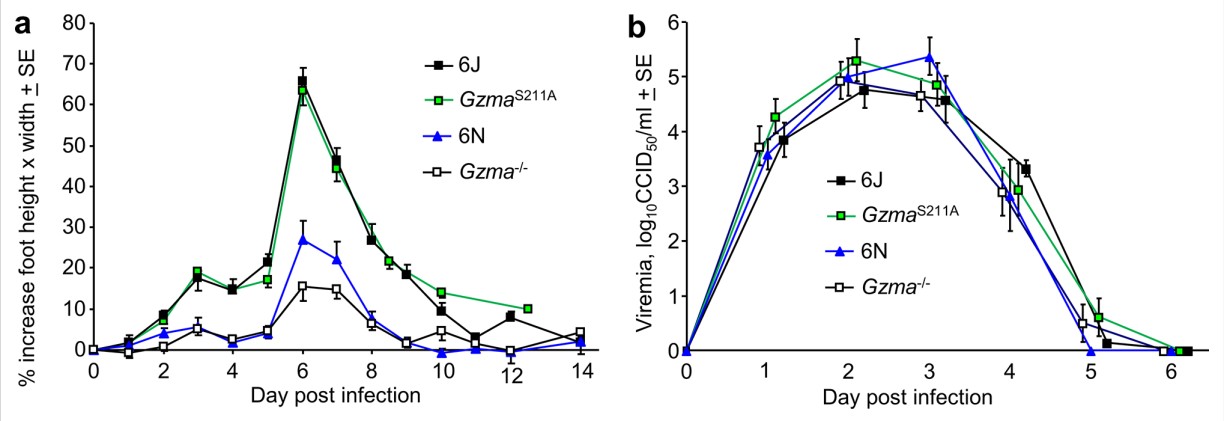

**Figure 1.** Chikungunya virus (CHIKV) infection in *Gzma*[-/-], *Gzma*[S211A], 6N, and 6J mice. (**a**) Percent increase in foot swelling for the indicated mouse strains. Data is from 2 to 4 independent experiments with 5–6 mice (10–12 feet) per group per experiment. From day 3 to day 10, feet from *Gzma*[S211A] and 6J mice were significantly more swollen than feet from *Gzma*[-/-] and 6N mice (Kolmogorov–Smirnov tests, p<0.002). (**b**) Viremia for the mice in (**a**) (6N n = 12, *Gzma*[S211A] n = 18, *Gzma*[-/-] n = 15, 6J n = 17).

The online version of this article includes the following figure supplement(s) for figure 1:

**Figure supplement 1.** Construction and characterization of *Gzma*[S211A] mice.

## Results

### CHIKV inflammatory arthritis in *Gzma*[S211A] mice

We reported previously that the inflammatory arthritis induced by CHIKV infection (manifesting as overt foot swelling) was significantly lower in *Gzma*[-/-] mice than in C57BL/6J (6J) mice (*Wilson et al., 2017*); an observation we confirm herein (*Figure 1a*). Injection of proteolytically active, but not proteolytically inactive, recombinant mouse GZMA induced inflammatory foot swelling, illustrating directly that GZMA's protease activity is able to drive pro-inflammatory responses (*Schanoski et al., 2019*). To confirm and extend these findings, a new homozygous GZMA active site mutant mouse was generated using CRISPR technology in 6J mice, with the reactive site serine changed to alanine (*Gzma*[S211A]) (*Susanto et al., 2013*; *Figure 1—figure supplement 1a–c*). Loss of enzyme activity was confirmed by BLT assays (*Suhrbier et al., 1991*; *Figure 1—figure supplement 1d*). Intracellular staining (*Schanoski et al., 2019*) showed that expression of the GZMA proteins in resting splenic NK cells (*Fehniger et al., 2007*) was comparable for *Gzma*[S211A] and 6J mice (*Figure 1—figure supplement 1e*).

When infected with CHIKV, viral titers in feet were not significantly different for *Gzma*[S211A] and 6J mice (*Figure 1—figure supplement 1f*), arguing that enzymically active GZMA has no significant antiviral activity against CHIKV. This is consistent with our previous study using *Gzma*[-/-] mice that also concluded that GZMA has no important antiviral function (*Wilson et al., 2017*). Surprisingly, however, foot swelling following CHIKV infection was *not* significantly different between the *Gzma*[S211A] and 6J mice (*Figure 1a*). If GZMA's pro-inflammatory bioactivity (*Wilson et al., 2017*) depends on its protease activity (*Schanoski et al., 2019*), the results from *Gzma*[-/-] and *Gzma*[S211A] mice (*Figure 1a*) would appear to provide contradictory results.

The effective amelioration of CHIKV arthritic foot swelling in 6J mice following treatment with Serpinb6b (an inhibitor of GZMA) also supported the view that GZMA promotes inflammation in this setting (*Wilson et al., 2017*). However, Serpinb6b also inhibited CHIKV foot swelling in *Gzma*[S211A] mice (*Figure 1—figure supplement 1g*), arguing that Serpinb6b can inhibit other unknown proteases involved in promoting arthritic inflammation. The contention is supported by the broad inhibitory activity of the human orthologue, SerpinB6 (*Strik et al., 2004*).

Taken together, these new data argued that the role of GZMA in promoting CHIKV arthritis and the results obtained from *Gzma*[-/-] mice (*Wilson et al., 2017*) required re-evaluation.

### *Gzma*[-/-] mice have a mixed C57BL/6N-C57BL/6J genetic background

To reconcile the apparent contradictory data from *Gzma*[S211A] and *Gzma*[-/-] mice, and cognizant of previously described issues with knockout mice (*Teoh et al., 2014*), whole-genome sequencing (WGS) of

*Gzma*[-/-] mice was undertaken (NCBI SRA; PRJNA664888). This analysis unequivocally demonstrated that *Gzma*[-/-] mice have a mixed genetic background, with ≈60% of the genome showing single-nucleotide polymorphisms (SNPs) and indels present in the C57BL/6N (6N) genome (*Mekada et al., 2015*; *Simon, 2013*), with the rest reflecting a 6J background (*Figure 3—figure supplement 1*). The strain origin of the BL/6-III ES cells used to generate *Gzma*[-/-] mice was only reported as C57BL/6 (*Ebnet et al., 1995*). As ES cells from 6J mice have a low rate of germline transmission, ES from 6N mice cells were frequently used to generate knockout mice (*Fontaine and Davis, 2016*), with our studies clearly arguing that BL/6-III ES cells also have a 6N background. The reported backcrossing of *Gzma*[-/-] mice onto C57BL/6 mice (*Müllbacher et al., 1999*) also clearly did not involve extensive backcrossing onto 6J mice.

Importantly, a large body of literature has resulted from the use of inbred *Gzma*[-/-] mice (and *Gzma*[-/-]/*Gzmb*[-/-] double knockout mice derived from them) with 6J mice used as controls, without being aware of the presence and potential confounding influence of the 6N background (*Supplementary file 1*).

## *Gzma*[-/-] mice and 6N (*Gzma*[+/+]) mice both have reduced foot swelling after CHIKV infection

An explanation for the significantly lower CHIKV-induced foot swelling seen in *Gzma*[-/-] mice is that the partial 6N background is influencing the foot swelling phenotype. To test this contention, 6J and 6N mice (both *Gzma*[+/+]) were infected with CHIKV. 6N mice showed a significant reduction in foot swelling when compared with 6J mice (*Figure 1a*), arguing that the 6N background, rather than loss of GZMA expression, was primarily responsible for amelioration of foot swelling in *Gzma*[-/-] mice.

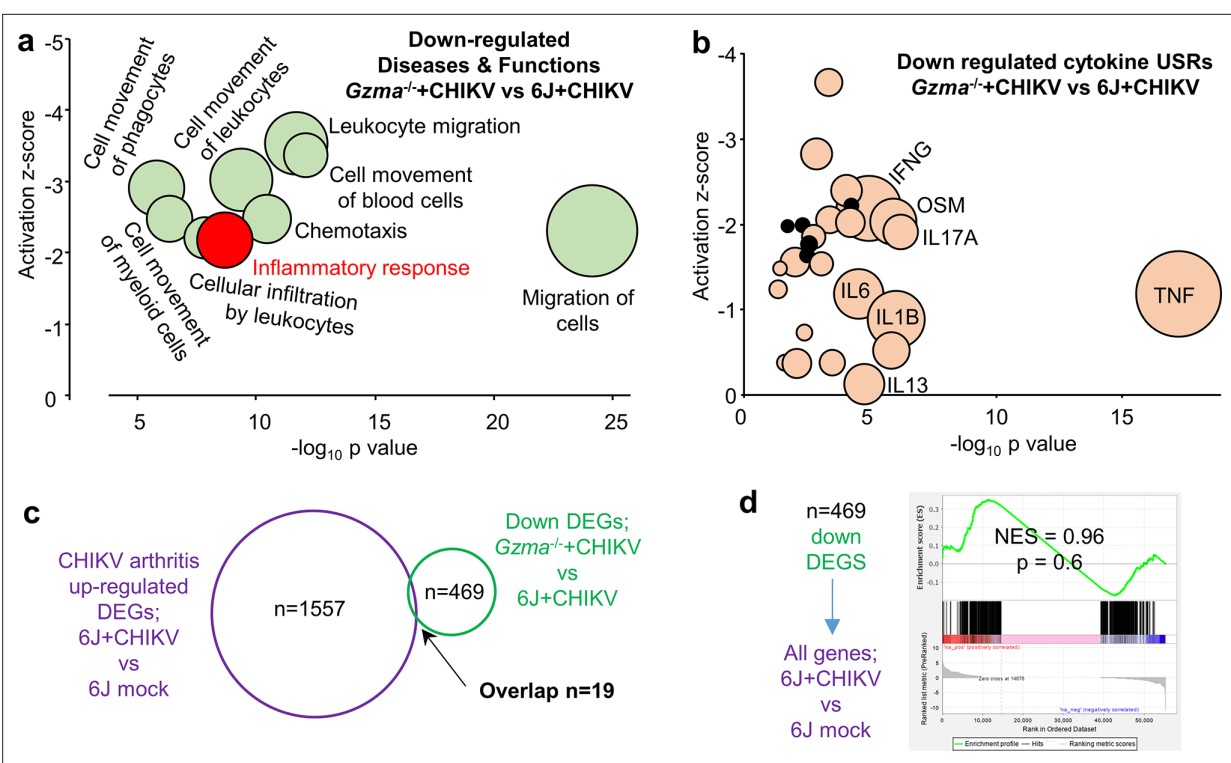

**Figure 2.** RNA-Seq of *Gzma*[-/-] + chikungunya virus (CHIKV) vs. 6J + CHIKV day 6 feet. (**a**) Selected Ingenuity Pathway Analysis (IPA) *Diseases and Function* annotations for the 469 downregulated differentially expressed genes (DEGs) for *Gzma*[-/-] + CHIKV vs. 6J + CHIKV (full set of annotations shown in *Supplementary file 2c*). (**b**) IPA cytokine upstream regulators (USRs) downregulated in CHIKV-infected feet of *Gzma*[-/-] mice (*Gzma*[-/-]+CHIKV vs. 6J + CHIKV; *Supplementary file 2e*) plotted by p-value and z-score. Black circles – minor USRs for *Gzma*[-/-] + CHIKV vs. 6J + CHIKV not identified for 6J + CHIKV vs. 6J mock (see also *Figure 4e*). (**c**) RNA-Seq identified 1557 DEGs upregulated in feet for 6J + CHIKV vs. 6J mock infection (*Supplementary file 2g*). RNA-Seq of *Gzma*[-/-] + CHIKV vs. 6J + CHIKV day 6 feet identified 469 downregulated DEGs in *Gzma*[-/-] mice associated with the reduced foot swelling (*Supplementary file 2b*). Only 19 of these DEGs were shared by these datasets. (**d**) Gene Set Enrichment Analysis (GSEA) of downregulated DEGs from *Gzma*[-/-] + CHIKV vs. 6J + CHIKV day 6 feet (*Supplementary file 2b*) vs. all genes (preranked by fold change) from feet 6J + CHIKV vs. 6J mock infection (*Supplementary file 2f*).

The reduced foot swelling in 6N and *Gzma*[-/-] mice was unlikely to be due to reduced viral loads as there were no significant differences in viremia for 6N, 6J, *Gzma*[-/-], or *Gzma*[S211A] mice (*Figure 1b*).

## RNA-Seq of CHIKV foot swelling for *Gzma*[-/-] vs. 6J mice

To gain insights into the reduced foot swelling in *Gzma*[-/-] mice (*Figure 1a*), RNA-Seq was undertaken on day 6 (peak arthritis) to compare gene expression in feet from CHIKV-infected *Gzma*[-/-] mice vs. 6J mice (NCBI BioProject PRJNA664644; full gene list in *Supplementary file 2a*). Differentially expressed genes (DEGs) (n = 1073) were generated after application of a q < 0.01 filter (*Supplementary file 2b*). When the 1073 DEGs (for *Gzma*[-/-] + CHIKV vs. 6J + CHIKV; *Supplementary file 2b*) were analyzed for *Diseases and Functions* using Ingenuity Pathway Analysis (IPA), the dominant annotations were associated with decreased cell movement (often leukocyte migration) (*Figure 2a*, *Supplementary file 2c*). These findings are consistent with immunohistochemistry data showing significantly reduced T cell and NK cells in the arthritic infiltrates in feet after CHIKV infection of *Gzma*[-/-] mice when compared with infected 6J mice (*Wilson et al., 2017*). IPA upstream regulator (USR) analysis of the 1073 DEGs (*Supplementary file 2d*) also illustrated that a series of pro-inflammatory cytokine USRs were down-regulated in the infected feet of *Gzma*[-/-] + CHIKV vs. 6J + CHIKV mice (*Figure 2b*, *Supplementary file 2e*), consistent with the reduced foot swelling seen in infected *Gzma*[-/-] mice when compared with infected 6J mice (*Wilson et al., 2017*).

We previously characterized the CHIKV arthritis signature by undertaking RNA-Seq of infected feet during peak foot swelling relative to control uninfected feet (6J + CHIKV vs. 6J mock infection) (*Wilson et al., 2017*). We reanalyzed the data (FastQ files NCBI BioProject PRJNA431476) using STAR aligner, RSEM EdgeR, and the more recent mouse genome build (GRCm38 Gencode vM23); all genes are shown in *Supplementary file 2f*, and a DEG list (with filters q < 0.01, fold change >2) is shown in *Supplementary file 2g*. The latter 2201 DEGs (for 6J + CHIKV vs. 6J mock infection) were analyzed by IPA, with the USRs shown in *Supplementary file 2h*. Of these USRs, 103 cytokine USR annotations showed significant upregulation (positive z-scores) after CHIKV infection (*Supplementary file 2i*). As might be expected, key pro-inflammatory cytokine USRs such as TNF, IFNG, IL6, and IL1B (*Suhrbier, 2019*) that were upregulated during CHIKV arthritis (*Supplementary file 2i*) were downregulated (negative z-scores) in the ameliorated foot swelling seen in *Gzma*[-/-] mice (*Figure 2b*, *Supplementary file 2e*). Curiously, however, five minor USRs (PPBP, CTF1, WNT7A, NAMPT, TIMP1) were downregulated in *Gzma*[-/-] mice, but were not upregulated during CHIKV infection (*Figure 2b*, black circles; *Supplementary file 2e*, yellow). Furthermore, of the 469 DEGs downregulated during CHIKV arthritis in *Gzma*[-/-] mice (*Supplementary file 2b*), only 19 were upregulated DEGs for CHIKV arthritis (*Figure 2c*, *Supplementary file 2g*). Gene Set Enrichment Analysis (GSEA) similarly revealed that DEGs downregulated in *Gzma*[-/-] mice were not significantly enriched in the upregulated genes for CHIKV arthritis in 6J mice (*Figure 2d*). Thus, although feet from CHIKV-infected *Gzma*[-/-] mice showed annotations associated with reduced cellular infiltrates and pro-inflammatory cytokines, this amelioration of arthritic signatures was associated with downregulation of genes largely not associated with CHIKV arthritis in 6J mice. In summary, arthritis amelioration in *Gzma*[-/-] mice was due to the downregulation of a largely distinct set of genes (and some distinct pathways), again arguing that the 6N background plays a key role in this phenotype.

## *Gzma*[-/-] mice have an intact nicotinamide nucleotide transhydrogenase gene

There are multiple genes associated with inflammation and/or arthritis that differ between *Gzma*[-/-] (mixed 6N/6J background) and 6J mice (*Supplementary file 3*). One gene that has been highlighted as a key difference between 6J and 6N mice is *Nnt* (*Freeman et al., 2006*; *Mekada et al., 2009*; *Ripoll et al., 2012*; *Ronchi et al., 2013*; *Vozenilek et al., 2018*; *Figure 3—figure supplement 1c*, red box). The function of NNT is primarily to sustain mitochondrial antioxidant capacity through the generation of NADPH, which supports the antioxidant capacity of the glutathione and thioredoxin systems (*McCambridge et al., 2019*; *Meimaridou et al., 2018*; *Ronchi et al., 2013*; *Rydström, 2006*; *Ward et al., 2020*). These systems are generally viewed as having broad anti-inflammatory activities (*Ghezzi, 2021*; *Yodoi et al., 2017*).

6N mice have a full-length *Nnt* gene with 21 protein-coding exons, whereas 6J mice have an in-frame 5-exon deletion removing exons 7–11 (*Freeman et al., 2006*). Confusingly, the MM10 mouse

build numbers the *Nnt* exons differently and includes the noncoding exon 1 that is located before the ATG start site. According to this numbering (which is used herein), 6J mice have lost exons 8–12 of 22 exons. The *Nnt* gene is located only ≈6.2 megabases from the *Gzma* gene on mouse chromosome 13, so >30 backcrosses would be required to segregate these two loci (**Silver, 2008**; **Figure 3—figure supplement 2**). The close association of *Nnt* and *Gzma* genes also means *Gzma*[-/-]*Gzmb*[-/-] double-knockout mice (**Supplementary file 1**) would also very likely have full-length *Nnt*. Another *Gzma*[-/-] mouse generated using 129/SvJ ES cells (**Shresta et al., 1997**) would likely have the same issue as 129/SvJ mice also have a full-length *Nnt* gene.

Alignment of the WGS of *Gzma*[-/-] mice (PRJNA664888) to the standard 6J MM10 mouse genome build allowed identification of the neomycin cassette insertion site into the *Gzma* gene that was used to generate the *Gzma*[-/-] mice (**Ebnet et al., 1995**; **Figure 3a**). Curiously, this alignment shows a 12-nucleotide insertion in the 6J genome at the *Nnt* exon 8–12 deletion junction (**Figure 3b**). The 12 nucleotides are also absent in other 6N WGS data (**Figure 3—figure supplement 3a**), indicating this is not a unique feature of *Gzma*[-/-] mice. This insertion in 6J may have accompanied deletion of *Nnt* exons 8–12 during the generation of 6J mice (**Fontaine and Davis, 2016**; **Figure 3—figure supplement 3b**).

Although a 6N genome sequence is available, it is poorly annotated, hence the C3H/HeJ genome build was used for alignments as it also has a full-length *Nnt* gene (**Figure 3—figure supplement 3c**).

Alignment of WGS reads from *Gzma*[-/-] mice to the C3H/HeJ genome clearly showed that *Gzma*[-/-] mice had a full-length *Nnt* gene, whereas 6J mice had the expected ≈16 kb deletion (**Figure 3c**). The approach (**Figure 3c**) was further validated using other 6J and non-6J WGS submissions (**Figure 3—figure supplement 3d**).

Sashimi plots of RNA-Seq reads aligned to the C3H/HeJ build clearly illustrated that *Nnt* mRNA from 6J mice was missing exons 8–12, whereas in *Gzma*[-/-] mice the full-length *Nnt* mRNA was expressed (**Figure 3d**, top). Alignment to the 6J genome and viewed by Sashimi plot showed that exons 7 and 13 are linked in the *Nnt* mRNA from 6J mice, consistent with expression of a truncated *Gzma* mRNA species. In contrast, exons 7 and 13 are *not* linked in the *Nnt* mRNA from *Gzma*[-/-] mice (**Figure 3d**, blue arrow) as the mRNA, but not the MM10 genome build, contains exons 8–12.

These results were confirmed by RT-PCR using primers located on either side of the exon 8–12 deletion (**Huang et al., 2006**; **Figure 3e**). *Gzma*[-/-] mice have the longer 6N *Nnt* PCR product as this *Nnt* mRNA includes exons 8–12, whereas 6J mice, *Gzma*[S211A] (generated by CRISPR on a 6J background), and type I IFN receptor knockout (*Ifnar*[-/-]) mice (also on a 6J background **Swann et al., 2007**), all showed a shorter PCR product, consistent with the deletion of exons 8–12 in the *Nnt* mRNA (**Figure 3e**). In a separate RT-PCR run, *Gzmb*[-/-] mice (**Wilson et al., 2017**) were shown to be missing *Nnt* exons 8–12, consistent with a 6J background (**Figure 3e**).

## The *Nnt* deletion promotes CHIKV-induced foot swelling

To determine whether the *Nnt* gene deletion seen in 6J mice might be responsible for promoting the arthritic foot swelling in CHIKV-infected mice, we generated 6N[ΔNnt8-12] mice wherein exons 8–12 of *Nnt* were deleted from 6N mice using CRISPR (**Figure 4—figure supplement 1**). 6N[ΔNnt8-12] mice thus have the same deletion of *Nnt* exons as 6J mice. As before, CHIKV-induced foot swelling was significantly higher in 6J mice when compared with 6N mice (**Figure 4a**). Importantly, foot swelling in 6N[ΔNnt8-12] mice was significantly higher than in 6N mice, with the *Nnt* exon 8–12 deletion increasing foot swelling to levels comparable to those seen in 6J mice (**Figure 4a**). There were no significant differences in viremia between the mouse strains (**Figure 4b**). This data illustrates that the absence of a functional *Nnt* gene (6N[ΔNnt8-12]) can by itself promote overt foot swelling after CHIKV infection. The data also argues that the presence of functional *Nnt* gene in *Gzma*[-/-] mice likely contributes to the ameliorated foot swelling seen in *Gzma*[-/-] mice.

Histology and H&E staining of arthritic feet illustrated that the increased foot swelling seen in 6N[ΔNnt8-12] mice (compared with 6N mice) was due primarily to increased edema (**Figure 4c and d**), with no significant differences in cellular infiltrates (**Figure 4—figure supplement 2a and b**). Edema is a recognized feature of alphaviral arthritides and is well described in CHIKV mouse models (**Gardner et al., 2010**; **Poo et al., 2014**; **Prow et al., 2019**). Immunohistochemistry with anti-CD3 also showed no significant differences in T cell numbers in the inflammatory infiltrates (**Figure 4—figure supplement 2c and d**). To further characterize the role of *Nnt* in CHIKV arthritis, day 6 feet from 6N[ΔNnt8-12]

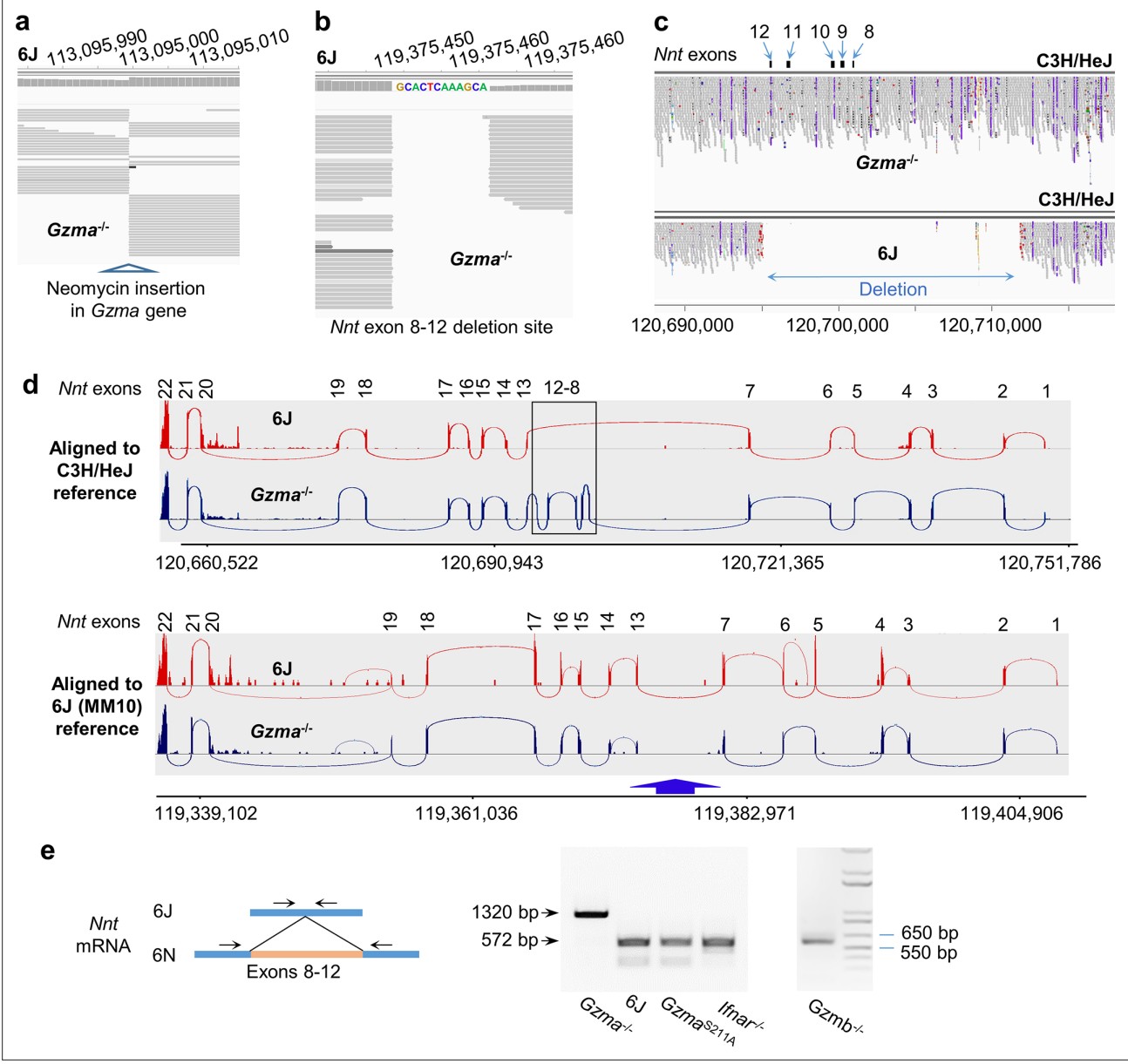

**Figure 3.** *Nnt* deletion in 6J but not *Gzma*⁻/⁻ mice. (**a**) Whole-genome sequencing (WGS) of *Gzma*⁻/⁻ mice aligned to the 6J (MM10) reference genome build, illustrating the insertion site of the neomycin cassette into the *Gzma* gene to create the knockout. (**b**) As for (**a**) but showing the site of the *Nnt* deletion, with the additional 12 nucleotides present in the 6J genome. (**c**) WGS of *Gzma*⁻/⁻ and 6J mice aligned to the C3H/HeJ reference genome, illustrating that the *Nnt* deletion present in 6J mice is absent in *Gzma*⁻/⁻ mice. The deletion is in chromosome 13; position 120,695,141–120,711,874 (C3H/HeJ numbering). (**d**) Reads from RNA-Seq of chikungunya virus (CHIKV)-infected 6J and *Gzma*⁻/⁻ mice aligned to the C3H/HeJ and 6J (MM10) reference genomes showing the Sashimi plot (Integrative Genomics Viewer) for the *Nnt* gene. (**e**) RT-PCR of testes using primers either side of exons 8–12 in the *Nnt* mRNA.

The online version of this article includes the following figure supplement(s) for figure 3:

**Figure supplement 1.** *Gzma*⁻/⁻ mice have a mixed 6N (yellow)/6J (green) background.

**Figure supplement 2.** *Nnt* and *Gzma* gene locations and backcrossing requirements.

**Figure supplement 3.** *Nnt* locus inserts and alignments.

+ CHIKV vs. 6N + CHIKV were compared using RNA-Seq (*Supplementary file 4a and b*). IPA USR analysis (*Supplementary file 4c*) provided 14 cytokine annotations with positive z-scores (*Supplementary file 4d*) that were associated with the increased foot swelling in 6N^ΔNnt8-12 mice (*Figure 4e*, brown circle). These 14 cytokine USRs were also upregulated during CHIKV arthritis (*Figure 4e*, yellow

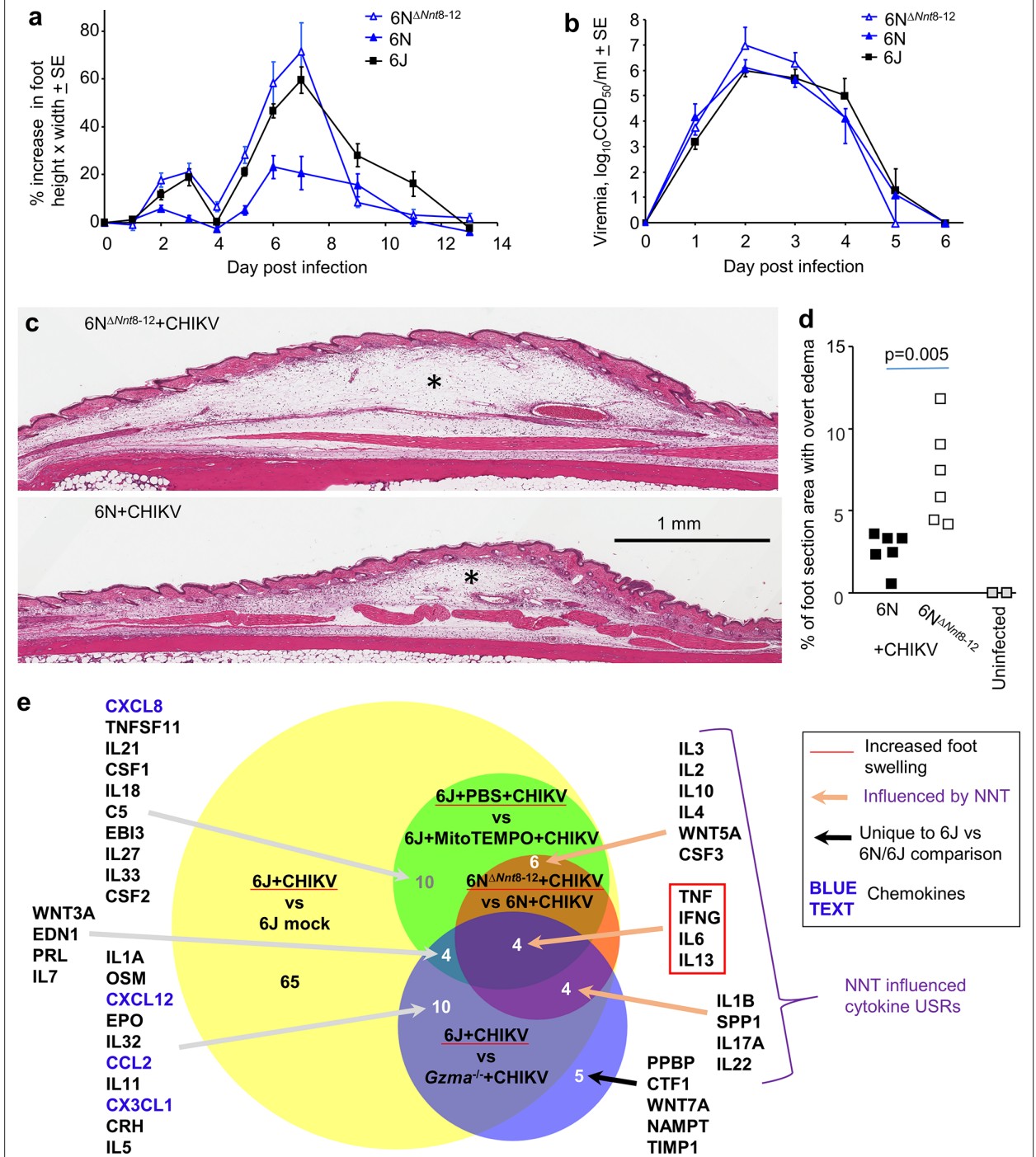

**Figure 4.** 6N$^{\Delta Nnt8\text{-}12}$ mice. (**a**) 6N$^{\Delta Nnt8\text{-}12}$ mice have the same *Nnt* exon deletion as 6J mice. Age-matched female 6N$^{\Delta Nnt8\text{-}12}$, 6N, and 6J mice were infected with chikungunya virus (CHIKV) and foot swelling measured over time (n = 5 mice and 10 feet per group). Foot swelling was significantly higher in 6N$^{\Delta Nnt8\text{-}12}$ mice when compared with 6N mice on days 2–7 (day 2 p=0.0026, day 7 p=0.0027, *t*-tests, parametric data distributions; days 3–6 p=0.003, Kolmogorov–Smirnov tests, nonparametric data distributions). Foot swelling was significantly lower in 6N mice when compared with 6J mice (day 2 p=0.042, day 6 p=0.001, day 7 p=0.0005, *t*-tests, parametric data distributions; days 3 and 5, p=0.002, Kolmogorov–Smirnov tests, nonparametric data distributions). (**b**) Viremia for the same mice as in (**a**). (**c**) H&E staining of feet from 6N$^{\Delta Nnt8\text{-}12}$ and 6N mice day 6 post infection showing subcutaneous edema (*). (**d**) Percentage of foot section area showing overt subcutaneous edema (statistics by Kolmogorov–Smirnov test). (**e**) RNA-Seq data for four comparisons was analyzed by Ingenuity Pathway Analysis (IPA) and cytokine upstream regulator (USR) overlaps shown. Only cytokine USRs with positive z-scores associated with increased foot swelling are shown.

The online version of this article includes the following figure supplement(s) for figure 4:

*Figure 4 continued on next page*

*Figure 4 continued*

**Figure supplement 1.** Generation of 6N$^{\Delta Nnt8-12}$ mice.

**Figure supplement 2.** Histological analyses for 6N$^{\Delta Nnt8-12}$ mice feet.

brown overlap). The IPA cytokine classification also includes chemokines, with no chemokine USRs identified within these 14 annotations (*Figure 4e*, brown circle), consistent with the lack of a significant cell migration phenotype (*Figure 4—figure supplement 2*). In contrast, several chemokine annotations were identified for the *Gzma*$^{-/-}$ mice (*Figure 4e*, blue circle, blue text) consistent with the reduced inflammatory infiltrate (*Wilson et al., 2017*). Furthermore, only 8/27 cytokine USRs identified for *Gzma*$^{-/-}$ + CHIKV vs. 6J + CHIKV were also identified for the 6N$^{\Delta Nnt8-12}$ + CHIKV vs. 6N + CHIKV comparison (*Figure 4e*, blue-brown overlap), arguing that other 6N background genes apart from *Nnt* ameliorated inflammatory cytokine activity in the feet of CHIKV-infected *Gzma*$^{-/-}$ mice.

Perhaps consistent with the arthritis literature in general (*Kato, 2020*; *Ma and Xu, 2013*; *Ogata et al., 2019*; *Suhrbier, 2019*), pronounced foot swelling in all comparisons (including MitoTEMPO, see below) was associated with upregulation of TNF, IFNG, and IL-6 USRs (*Figure 4e*, red box). IL-13 was also a consistently upregulated USR, with IL-13 associated with resolution of arthritic inflammation (*Schett, 2019*), with peak CHIKV arthritis in 6J mice associated with a significant resolution phase signature (*Prow et al., 2019*).

## MitoTEMPO ameliorates CHIKV arthritis in 6J mice

As NNT's primary function is to sustain mitochondrial antioxidant capacity (*McCambridge et al., 2019*; *Meimaridou et al., 2018*; *Ronchi et al., 2013*; *Rydström, 2006*; *Ward et al., 2020*), the data argues that the reduced arthritic foot swelling in 6N mice (and to some extent in *Gzma*$^{-/-}$ mice) is due to increased mitochondrial antioxidant capacity. MitoTEMPO has been widely used as an experimental antioxidant treatment to scavenge mitochondrial ROS in a variety of disease settings (*Aoyama et al., 2012*; *Li et al., 2018*; *To et al., 2020*; *Vincent et al., 2020*; *Wu et al., 2020*). Treatment of CHIKV arthritis in 6J mice with MitoTEMPO from day 3 to 8 post infection significantly ameliorated peak foot swelling (*Figure 5a*). Viremia was not significantly affected by MitoTEMPO treatment, even when treatment was initiated on day 0 (*Figure 5b*).

Histology and H&E staining revealed that MitoTEMPO treatment significantly reduced the cellular infiltrate in feet day 6 post CHIKV infection of 6J mice when compared with PBS treatment (*Figure 5c and d*). Edema was also significantly reduced by MitoTEMPO treatment (*Figure 5e and f*); additional images are shown in *Figure 5—figure supplement 1*. RNA-Seq analysis (*Supplementary file 5a and b*) and IPA USR analysis (*Supplementary file 5c*) illustrated that MitoTEMPO treatment was associated with downregulation of 24 cytokine USRs (*Figure 4e*, green circle; *Supplementary file 5d*), with 10 of these also downregulated in 6N vs. 6N$^{\Delta Nnt8-12}$ mice (*Figure 4e*, green and brown circle overlap). MitoTEMPO treatment in 6J mice and an intact *Nnt* gene in 6N mice thus provided overlapping anti-inflammatory activities. However, MitoTEMPO was also able to inhibit a series of additional proinflammatory arthritic responses including the chemokine CXCL8 and complement factor 5 (C5), with complement promoting arthritic infiltrates in a related alphavirus, Ross River virus (*Morrison et al., 2007*). Inhibition of CXCL8 and/or C5 is consistent with the reduced cellular infiltrate (*Figure 5c and d*).

## Reinvestigation of the physiological role of GZMA using poly(I:C)

Our current understanding of the physiological function of GZMA comes, to a large extent, from multiple studies comparing *Gzma*$^{-/-}$ (and *Gzma*$^{-/-}$/*Gzmb*$^{-/-}$) mice with 6J mice (*Supplementary file 1*). Given the results herein, many of the reported phenotypes are likely to have arisen, at least in part, from an intact *Nnt* gene and/or the 6N background, complicating any conclusions regarding the physiological role of GZMA.

To gain new insights into GZMA's activity in vivo, we sought to find an experimental setting where high levels of GZMA are secreted. We have shown previously that humans, nonhuman primates, and mice have elevated serum GZMA levels after infection with CHIKV (*Schanoski et al., 2019*; *Wilson et al., 2017*). Infection of mice with a series of other RNA viruses also resulted in high serum GZMA levels early in infection, with NK cells identified as the likely source (*Schanoski et al., 2019*). Resting

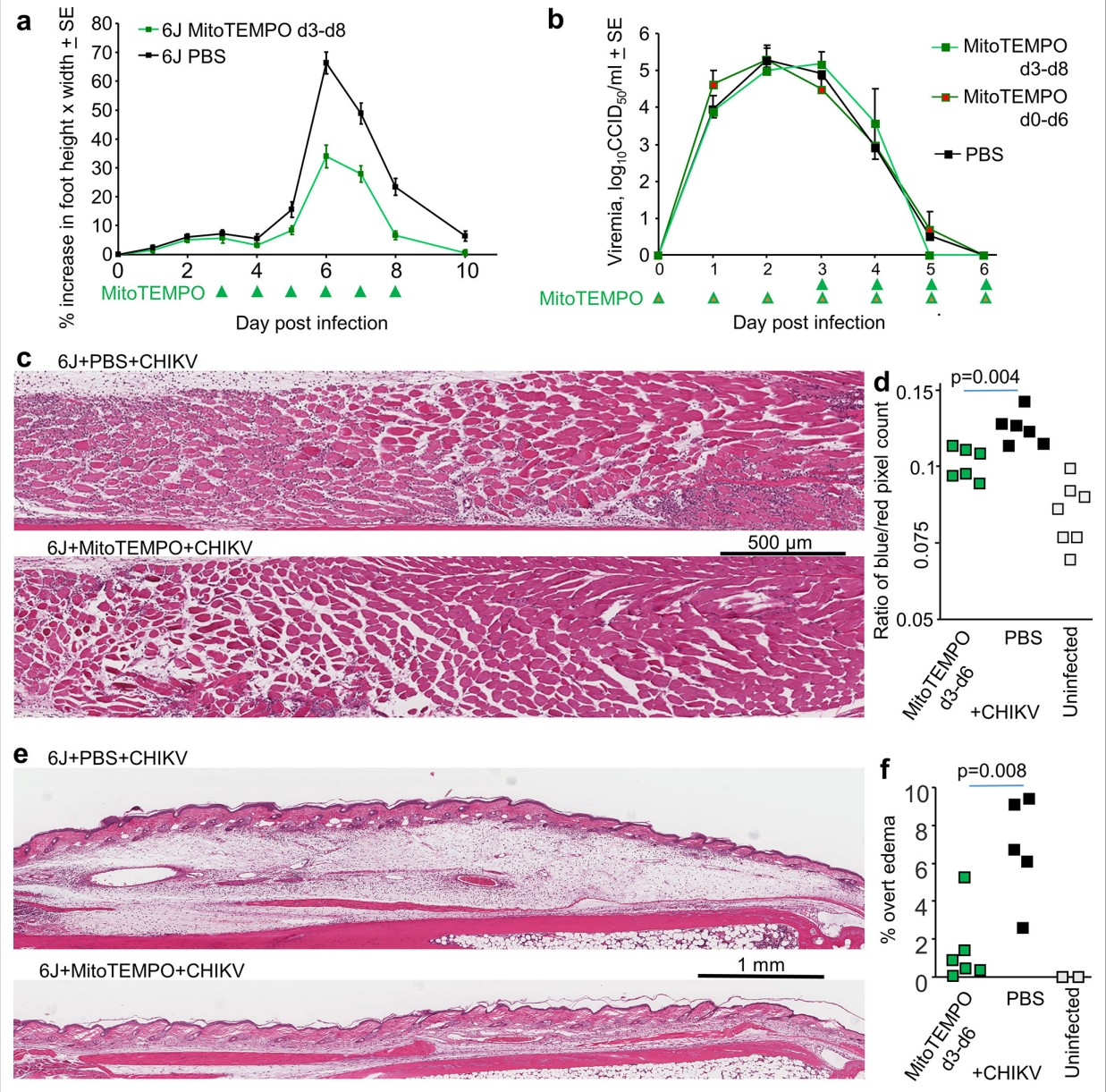

**Figure 5.** MitoTEMPO treatment. (**a**) 6J mice were infected with chikungunya virus (CHIKV) and then treated with MitoTEMPO or PBS i.v. daily on days 3–8 post infection and foot swelling measured (n = 5/6 mice and 10/12 feet per group). Statistics by *t*-tests, days 6, 7, and 8, *p* < 0.001. (**b**) Viremia for the mice in (**a**), with an additional group of six mice treated daily with MitoTEMPO from day 0 till day 6. (**c**) H&E staining of muscle tissues in feet of mice day 6 post CHIKV infection for mice treated with PBS or MitoTEMPO from day 3 to day 6. Clusters of small blue spots in and around the muscle bundles illustrate nuclei of infiltrating leukocytes. (**d**) Aperio pixel count of blue (nuclear) over red (cytoplasmic) pixels; a measure of leukocyte infiltration (statistics by *t*-test). (**e**) H&E staining of feet as in (**c**) but showing overt subcutaneous edema. (**f**) Percentage of foot section areas showing overt subcutaneous edema (statistics by *t*-test).

The online version of this article includes the following figure supplement(s) for figure 5:

**Figure supplement 1.** Additional H&E images of feet subcutaneous edema (as in *Figure 5e*).

NK cells constitutively contain abundant levels of GZMA protein (*Fehniger et al., 2007*), which is usually stored in granules as a mature protease, with the low pH of the granule preventing (premature) proteolytic activity (*Stewart et al., 2012*).

Polyinosinic:polycytidylic acid (poly(I:C)) can often mimic aspects of the innate responses to RNA viruses (*Prow et al., 2017*). We thus injected poly(I:C) i.v. into 6J mice and showed that serum GZMA levels reached mean peak serum levels of ≈20 ng/ml of serum after 2 hr, with levels dropping to

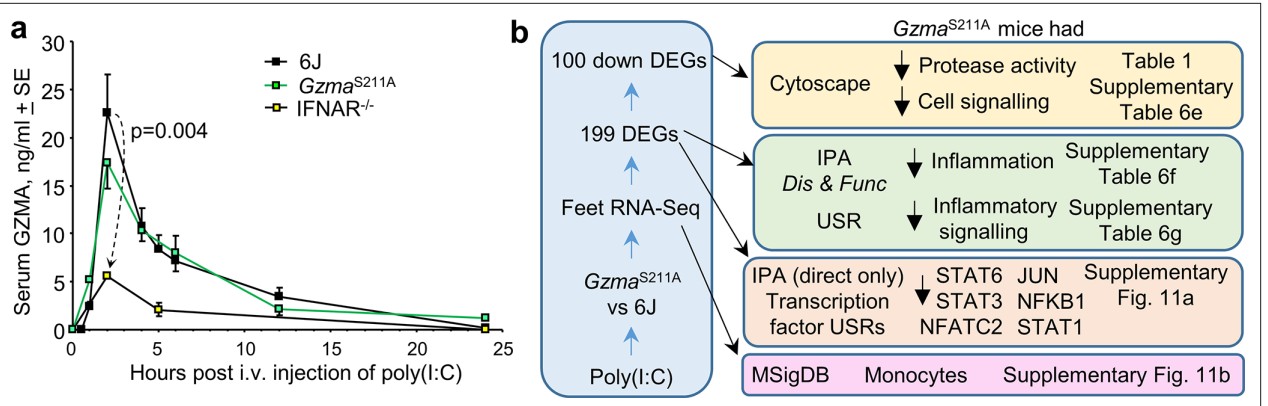

**Figure 6.** Polyinosinic:polycytidylic acid (poly(I:C)) injection into *Gzma*^S211A, 6J mice, and *Ifnar*^-/-. (**a**) *Gzma*^S211A, 6J, and *Ifnar*^-/- mice were injected i.v. with 250 µg of poly(I:C) in 150 µl of PBS, and serum samples were taken at the indicated times and assayed for GZMA concentration using a capture ELISA kit (6J n = 5–8, *Ifnar*^-/- n = 5–6 and *Gzma*^S211A n = 3 mice per time point). (**b**) *Gzma*^S211A and 6J mice were injected i.v. with 250 µg of poly(I:C) and feet removed 6 hr later and analyzed by RNA-Seq. The differentially expressed genes (DEGs) (**Supplementary file 6b**) were analyzed by Cytoscape and Ingenuity Pathway Analysis (IPA). The full gene list (**Supplementary file 6a**) was analyzed using the Molecular Signature Database (MSigDB).

The online version of this article includes the following figure supplement(s) for figure 6:

**Figure supplement 1.** Serum granzyme B (GZMB) levels after i.v. injection of polyinosinic:polycytidylic acid (poly(I:C)).

**Figure supplement 2.** RNA-Seq for polyinosinic:polycytidylic acid (poly(I:C)) injection into *Gzma*^S211A vs. 6J mice.

**Figure supplement 3.** The gene expression signatures in feet seen 6 hr after injection of polyinosinic:polycytidylic acid (poly(I:C)) in *Gzma*^S211A vs. 6J mice are not significantly recapitulated in spleen.

**Figure supplement 4.** Ingenuity Pathway Analysis (IPA) and Gene Set Enrichment Analysis (GSEA) for polyinosinic:polycytidylic acid (poly(I:C)) injection into *Gzma*^S211A vs. 6J mice.

baseline after 24 hr (**Figure 6a**). Although poly(I:C) treatment is known to activate NK cells (**Djeu et al., 1979**; **Fehniger et al., 2007**; **Miyake et al., 2009**; **Ngoi et al., 2008**), this rapid and prodigious poly(I:C)-induced release of GZMA into the circulation, to the best of our knowledge, has hitherto not been reported. Type I interferons (IFNs) are also rapidly induced by poly(I:C) (**Dempoya et al., 2012**; **Santiago-Raber et al., 2003**), and NK cells express the type I IFN receptor and can respond to type I IFNs (**Madera et al., 2016**; **Mizutani et al., 2012**). Injection of poly(I:C) i.v. into *Ifnar*^-/- mice resulted in a significantly blunted elevation of GZMA (**Figure 6a**). Type I IFNs thus appear to augment this rapid GZMA release; however, the absence of type I IFN signaling does not prevent GZMA secretion, consistent with previous data (**Schanoski et al., 2019**). Poly(I:C) treatment of *Gzma*^S211A mice resulted in serum GZMA levels not significantly different from those seen in 6J mice (**Figure 6a**), illustrating that the active site mutation does not significantly affect production, secretion, or stability. Finally, although GZMB and GZMA are often considered to be co-expressed (**Supplementary file 1**), in this setting no serum GZMB was detected (**Figure 6—figure supplement 1**). GZMB and perforin proteins are not expressed in resting NK cells and appear only after ≈24 hr of appropriate stimulation (**Fehniger et al., 2007**).

## RNA-Seq after poly(I:C) injection in *Gzma*^S211A vs. 6J mice

*Gzma*^S211A and 6J mice are both on a 6J genetic background, are both missing *Nnt* exons 8–12, and both show similar levels of serum GZMA secretion after poly(I:C) treatment (**Figure 6a**). Thus, the only difference between these strains is that GZMA in *Gzma*^S211A mice is enzymically inactive (**Figure 1—figure supplement 1d**). *Gzma*^S211A and 6J mice were injected with poly(I:C) (as in **Figure 6a**), with feet and spleens removed 6 hr later and analyzed by RNA-Seq (NCBI BioProject PRJNA666748). The rationale for this time point was to capture early transcriptional events after the peak of serum GZMA. The sample preparation strategy and RNA-Seq data overview is provided in **Figure 6—figure supplement 2**. The full gene list for feet is provided in **Supplementary file 6a**, with the 199 DEGs listed in **Supplementary file 6b**. For spleen, the full gene list is shown in **Supplementary file 6c** and the four DEGs in **Supplementary file 6d**. This represents the first study of GMO mice targeting *Gzma* that is free from the potentially confounding influence of the mixed 6J/6N background.

**Table 1.** Cytoscape analysis of downregulated differentially expressed genes (DEGs) in *Gzma*[S211A] mice.

RNA-Seq of feet taken from *Gzma*[S211A] vs. 6J mice 6 hr after polyinosinic:polycytidylic acid (poly(I:C)) injection provided 199 differentially expressed genes (DEGs), of which 100 were downregulated in *Gzma*[S211A] mice (**Supplementary file 6b**). When analyzed by Cytoscape, the top annotations were associated with negative regulation (underlined) of protease activities (bold) or negative regulation of protein metabolism (which includes anabolism and catabolism). Also significant were a series of annotations associated with negative regulation of cell signaling (italics). The complete list is shown in **Supplementary file 6e**; top annotations are shown here with duplicates removed.

| Category | Description | FDR value |
|---|---|---|
| GO Process | <u>Negative</u> regulation of cellular protein metabolic process | 1.91E-06 |
| UniProt Keywords | **Protease inhibitor** | 9.20E-06 |
| SMART Domains | **SERine Proteinase INhibitors** | 2.60E-05 |
| GO Process | Negative **regulation of catalytic activity** | 6.73E-05 |
| GO Process | <u>Negative</u> regulation of nitrogen compound metabolic process | 7.86E-05 |
| GO Process | <u>Negative</u> regulation of cellular metabolic process | 7.86E-05 |
| InterPro Domains | **Serpin superfamily** | 1.10E-04 |
| GO Process | <u>Negative</u> regulation of molecular function | 1.30E-04 |
| GO Function | **Enzyme inhibitor activity** | 1.50E-04 |
| UniProt Keywords | **Serine protease inhibitor** | 1.50E-04 |
| GO Process | <u>Negative</u> regulation of macromolecule metabolic process | 1.90E-04 |
| GO Process | <u>Negative</u> regulation of protein modification process | 2.60E-04 |
| GO Process | Negative **regulation of hydrolase activity** | 2.80E-04 |
| GO Function | **Serine-type endopeptidase inhibitor activity** | 6.60E-04 |
| GO Process | <u>Negative</u> regulation of phosphate metabolic process | 7.40E-04 |
| GO Function | **Endopeptidase inhibitor activity** | 8.50E-04 |
| GO Process | Negative **regulation of endopeptidase activity** | 0.001 |
| GO Process | Negative *regulation of intracellular signal transduction* | 0.0014 |
| GO Process | Negative *regulation of protein phosphorylation* | 0.0014 |
| GO Process | Regulation of protein metabolic process | 0.0015 |
| GO Process | Negative *regulation of MAPK cascade* | 0.0061 |
| GO Process | Negative *regulation of cellular process* | 0.0063 |
| GO Process | Negative *regulation of biological process* | 0.0071 |

## Immune/inflammation signatures stimulated by circulating proteolytically active GZMA

Circulating proteolytically active GZMA would appear to have limited influence in the spleen as only four DEGs were identified in the spleen of poly(I:C)-treated *Gzma*[S211A] vs. 6J mice (**Supplementary file 6d**). Interestingly, the only significantly upregulated DEG in *Gzma*[S211A] spleens was Mid1, a gene involved in controlling granule exocytosis by cytotoxic lymphocytes (**Boding et al., 2014**; **Boding et al., 2015**). GSEA also illustrated that neither up- nor downregulated DEGs identified in the feet were enriched in spleen (**Figure 6—figure supplement 3**), arguing that in this setting the activity of GZMA in the periphery is not significantly recapitulated in spleen.

Of the 199 DEGs identified in the feet of poly(I:C)-treated *Gzma*[S211A] vs. 6J mice, 100 were downregulated, with the top annotations associated with negative regulation of protease activity and negative regulation of cell signaling after Cytoscape analysis (**Figure 6b**, **Table 1**, **Supplementary file 6**).

This result supports the view that GZMA is proteolytically active in vivo (*Spaeny-Dekking et al., 2000*) and that GZMA's proteolytic activity mediates cell signaling events under physiological conditions. IPA *Disease and Functions* analysis of the 199 DEGs from feet (*Gzma*^S211A + polyIC vs. 6J + polyIC; *Supplementary file 6b*) identified downregulation (negative z-scores) of a series of inflammation and leukocyte activation signatures (*Figure 6b*, *Supplementary file 6f*). IPA USR analysis (core analysis with direct and indirect interactions) indicated downregulation of a series of cytokine, immune receptor, and transcription factor USRs (*Figure 6b*, *Supplementary file 6g*). An IPA USR analysis using the direct-only interaction option, which largely limits the analysis to transcription factors, showed STAT6, STAT3, NFATC2, JUN, NFKB1, and STAT1 to be the dominant downregulated transcription factor signatures in *Gzma*^S211A mice by z-score and p-values (*Figure 6b*, *Figure 6—figure supplement 4a*). These transcription factors are associated with various innate and adaptive immune responses, with NFATC2 playing a central role in the activation of T cells during the development of an immune response. STAT3 and NF-κB have previously been shown to be activated in macrophages by recombinant GZMA in vitro (*Santiago et al., 2020*), with monocytes/macrophages reported as targets for GZMA activity in a variety of settings (*Garzón-Tituaña et al., 2021*; *Metkar et al., 2008*; *Santiago et al., 2017*; *Spencer et al., 2013*; *Uranga et al., 2016*). Interrogation of the Molecular Signature Database (MSigDB) (*Subramanian et al., 2005*) using GSEAs also identified gene sets associated with activated monocytes (GSE19888) that were significantly enriched in the downregulated genes for *Gzma*^S211A + polyIC vs. 6J + polyIC (*Figure 6—figure supplement 4b*). This observation supports the contention that monocytes/macrophages are activated by circulating GZMA (*Figure 6b*). IPA of the 150 core enriched genes from these GSEAs also identified STAT6, STAT3, NFATC2, JUN, NFKB1,

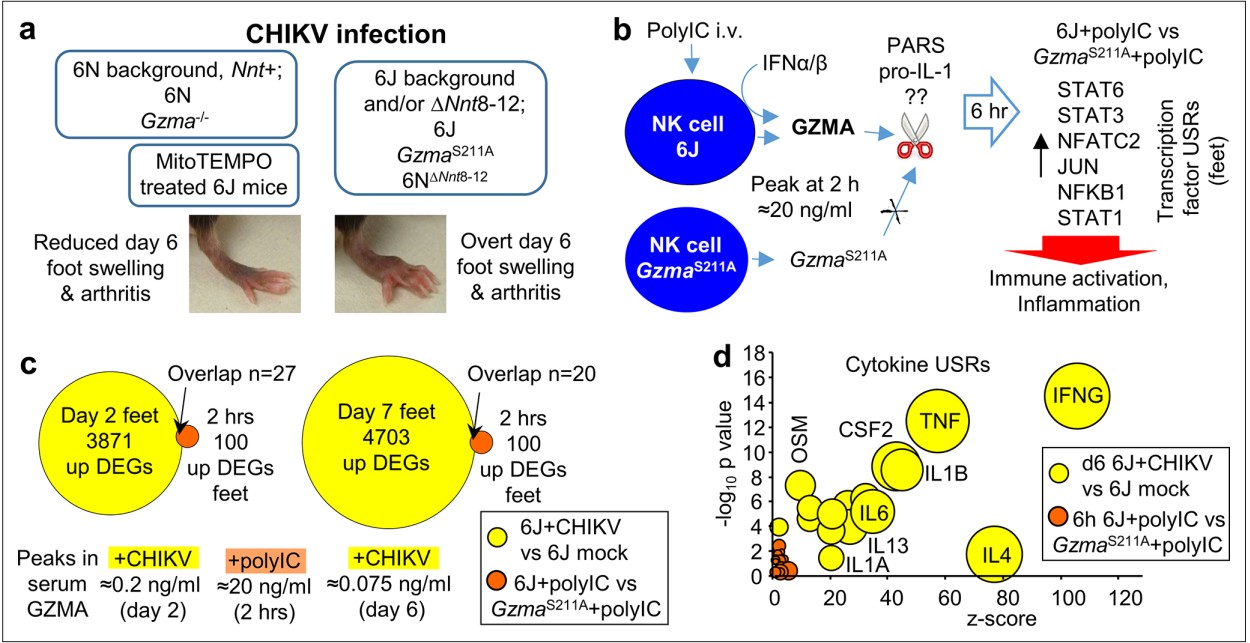

**Figure 7.** Summary of chikungunya virus (CHIKV) foot swelling and granzyme A (GZMA) bioactivity, and the poor concordance between the two. (**a**) Summary of foot swelling results. 6N and *Gzma*^-/- mice have a 6N or mixed 6J/6N background and have an intact *Nnt* gene and show reduced CHIKV-induced foot swelling. MitoTEMPO treatment also reduced foot swelling. 6J, *Gzma*^S211A, and 6N^ΔNnt8-12 mice are all missing exons 8–12 of *Nnt* and show increased foot swelling. (**b**) Summary of GZMA bioactivity. Using polyinosinic:polycytidylic acid (poly(I:C)) to induce high levels of GZMA secretion from NK cells, studies in *Gzma*^S211A mice illustrated that proteolytically active circulating GZMA promotes certain immune-stimulating/pro-inflammatory responses (dominant transcription factor upstream regulators (USRs) are shown; Figure S11a). No clear consensus has emerged regarding the molecular target(s) of GZMA (??); two potential extracellular candidate targets are shown; protease activated receptors and pro-IL-1. (**c**) Low overlap between CHIKV and GZMA induced differentially expressed genes (DEGs). DEGs upregulated in feet by CHIKV infection of 6J mice on days 2 and 7 (a low stringency filter of q < 0.05 was applied to the all gene lists in *Supplementary file 2f* to provide these DEGs) were compared with the DEGs upregulated in feet by active GZMA in 6J mice 6 hr after poly(I:C) treatment (i.e., downregulated in *Gzma*^S211A mice; *Supplementary file 6b*). Overlapping genes (n = 27 and 20) are listed in *Supplementary file 6i*. The mean peak levels of serum GZMA for each group are shown below the Venn diagrams. (**d**) Cytokine USRs for GZMA vs. CHIKV. A series of cytokine USRs were induced by proteolytically active GZMA (i.e., upregulated for 6J + polyIC vs. *Gzma*^S211A + polyIC; *Supplementary file 6g*, column R). The same USRs were significantly more upregulated by CHIKV infection (*Supplementary file 2i*). Data for GZMA (orange) and CHIKV (yellow) are plotted with bubble size representing number of molecules in dataset.

and STAT1 as significant USRs, even though only 10 of these 150 genes were significant DEGs for *Gzma*[S211A] vs. 6J (***Supplementary file 6h***). These USR signatures would thus appear to be a consistent feature of the RNA-Seq data for *Gzma*[S211A] + polyIC vs. 6J + polyIC.

## Summary of CHIKV foot swelling results and GZMA's protease bioactivity

The CHIKV foot swelling data so far is summarized in ***Figure 7a*** and argues that the 6N background, which includes a functional *Nnt* gene, rather than the absence of GZMA expression, causes the ameliorated foot swelling in *Gzma*[-/-] mice (***Figure 1a***). The presence of an intact *Nnt* gene can itself reduce foot swelling, although other 6N background genes in *Gzma*[-/-] mice also contribute (***Figure 4e***). MitoTEMPO treatment and NNT activity share anti-inflammatory activities (***Figure 4e***), presumably because both are involved in mitochondrial ROS mitigation. As CHIKV-infected *Gzma*[S211A] mice (on a 6J background) show no reduction in foot swelling, the data argues that GZMA is not a major player in CHIKV arthritis.

The *Gzma*[S211A] poly(I:C) data is summarized in ***Figure 7b***. Poly(I:C) induces high levels of circulating GZMA, which is potentiated by type I IFNs, with NK cells the likely source of GZMA (***Schanoski et al., 2019***). IPA of RNA-Seq data comparing *Gzma*[S211A] + poly(I:C) vs. 6J + poly(I:C) supports the view that GZMA's proteolytic function is required for its bioactivity (***Schanoski et al., 2019***; ***Figure 6b***, ***Table 1***). Circulating proteolytically active GZMA promotes certain immune-stimulating and pro-inflammatory activities (***Schanoski et al., 2019***; ***Shimizu et al., 2019***; ***van Daalen et al., 2020***; ***Wensink et al., 2015***), with STAT6, STAT3, NFATC2, JUN, NFKB1, and STAT1 identified as dominant transcription factor USRs (***Figures 1 and 7b***; ***Figure 6—figure supplement 4a***). Consensus regarding the molecular target(s) of extracellular GZMA's protease activity (***Figure 7b***, indicated as ??) remains elusive, but may include protease-activated receptors (***Hansen et al., 2005***; ***Sower et al., 1996***; ***Suidan et al., 1994***; ***Suidan et al., 1996***) and/or pro-IL-1 ***Hildebrand et al., 2014***; the latter can become extracellular when cells lyse (***Afonina et al., 2015***).

## Minor role for GZMA in CHIKV infection and arthritis

If proteolytically active GZMA is present and has immune-stimulating and pro-inflammatory activities (***Figure 7b***), why does it have no significant role in driving the overt CHIKV arthritic foot swelling (with no ameliorated foot swelling in *Gzma*[S211A] mice, ***Figure 7a***)? Firstly, the serum GZMA levels during CHIKV infection and arthritis (≈0.2 and ≈0.075 ng/ml) were substantially lower than those seen after poly(I:C) treatment (≈20 ng/ml) (***Figure 7c***), and very much lower than the ≈5 µg of recombinant GZMA-injected intraplantar to generate overt foot swelling in the absence of any other stimuli (***Schanoski et al., 2019***). Secondly, when the DEGs (q < 0.05) that were upregulated in feet during CHIKV peak viremia (day 2) and peak arthritis (day 7 in the Wilson et al. study) (6J + CHIKV vs. 6J mock infection) were compared with DEGs upregulated by proteolytically active GZMA (6J + poly(I:C) vs. *Gzma*[S211A] + poly(I:C); ***Supplementary file 6b***), only small overlaps were evident, 27 genes for day 2 and 20 genes for day 7 (***Figure 7c***, ***Supplementary file 6i***). Thirdly, although CHIKV infection showed upregulation of many transcription factor and cytokine USRs (***Supplementary file 2h***), with some of these also upregulated by poly(I:C) treatment (***Supplementary file 6g***), the magnitude of the effects (by p-value and z-scores) was very much smaller for poly(I:C) treatment (shown for cytokine USRs, ***Figure 7d***). So CHIKV infection (day 2) and arthritis (day 6) stimulate immune and inflammation pathways that overlap with those stimulated by GZMA, but GZMA only plays a minor role in stimulating these pathways during CHIKV viremia (day 2) and arthritis (day 6).

## 6J SRA accessions with *Nnt* exon reads inconsistent with a 6J background

Given the data presented herein and elsewhere (***Bourdi et al., 2011***; ***Fontaine and Davis, 2016***; ***Leskov et al., 2017***; ***McCambridge et al., 2019***; ***Mekada et al., 2009***; ***Rao et al., 2020***; ***Ripoll et al., 2012***; ***Rydström, 2006***; ***Toye et al., 2005***; ***Vozenilek et al., 2018***) and given that redox regulation affects many cellular processes (***Gambhir et al., 2019***; ***Lingappan, 2018***; ***Sun et al., 2020***), *Nnt* emerges as a legitimate focus of concern. In addition, the presence of *Nnt* exons 8–12 provides a useful genetic marker to illustrate that a mouse strain is not on a pure 6J background, with genetic backgrounds, as shown herein and elsewhere, able to have a profound influence on phenotype

(*Leskov et al., 2017*; *Morales-Hernandez et al., 2018*; *Salerno et al., 2019*; *Vozenilek et al., 2018*; *Williams et al., 2021*; *Wolf et al., 2016*). We thus undertook a k-mer mining approach to interrogate the NCBI SRA (*Figure 8—figure supplement 1a*), which (at the time of analysis) contained 61,443 RNA-Seq Run Accessions listing 'C57BL/6J' in the strain field of the metadata.

For 'C57BL/6J' Run Accessions, k-mer mining was used to count the number of RNA-Seq reads with sequence homology to *Nnt* exon 2 or exon 9, with these two exons being of similar length (203 bp for exon 2 and 192 bp for exon 9). RNA-Seq analysis of 6J tissues would ordinarily provide reads for exon 2, whereas the presence of exon 9 reads would be inconsistent with a pure 6J background. A conservative k-mer mining approach was used; (i) only an exact match for 'C57BL/6J' in the strain field was allowed, (ii) Run Accessions with small or large compressed file sizes (<200 Mb and >1500 Mb) were excluded, (iii) nucleotide mismatches for the 31-nucleotide k-mers were disallowed, (iv) where there were technical replicates, only one was mined, and (v) a read count of ≥10 per exon was used as a cutoff. This k-mer mining analysis revealed that 1008 Run Accessions had reads aligning to both *Nnt* exons 2 and 9, indicating full-length *Nnt* (*Nnt+*), which is not consistent with a 6J background (*Supplementary file 7a*). In contrast, 2469 had exon 2 reads, but no exon 9 reads indicating truncated *Nnt* (*Nnt-*), which *is* consistent with a 6J background (*Supplementary file 7b*). Lastly, 267 Run Accessions had equivocal results (*Supplementary file 7c*). Therefore ≈27% (1008 of 3744) of Run Accessions listing 'C57BL/6J' in the strain field had sequencing reads not consistent with a 6J background. The k-mer mining approach was validated for a selected group of Run Accessions using NCBI BLAST alignments, which illustrated excellent concordance with the k-mer mining read count data (*Supplementary file 7d*).

The startlingly high percentage (≈27%) of SRA Accessions listing 'C57BL/6J' but having *Nnt* reads inconsistent with a 6J genetic background argues that the *Nnt* gene and associated mixed 6N/6J genetic backgrounds are widely underappreciated in a broad range of research areas. It should be noted that a large number of Run Accessions (n = 206,586) list 'C57BL/6' in the strain field and thus do not provide information on the substrain (*Mekada et al., 2009*) being used.

## BioProjects comparing mice with truncated *Nnt* to mice with full-length *Nnt*

Based on the results of k-mer mining of 'C57BL/6J' Run Accessions, BioProjects (n = 373) were grouped into three categories: (i) BioProjects where all the Run Accessions with 'C57BL/6J' in the

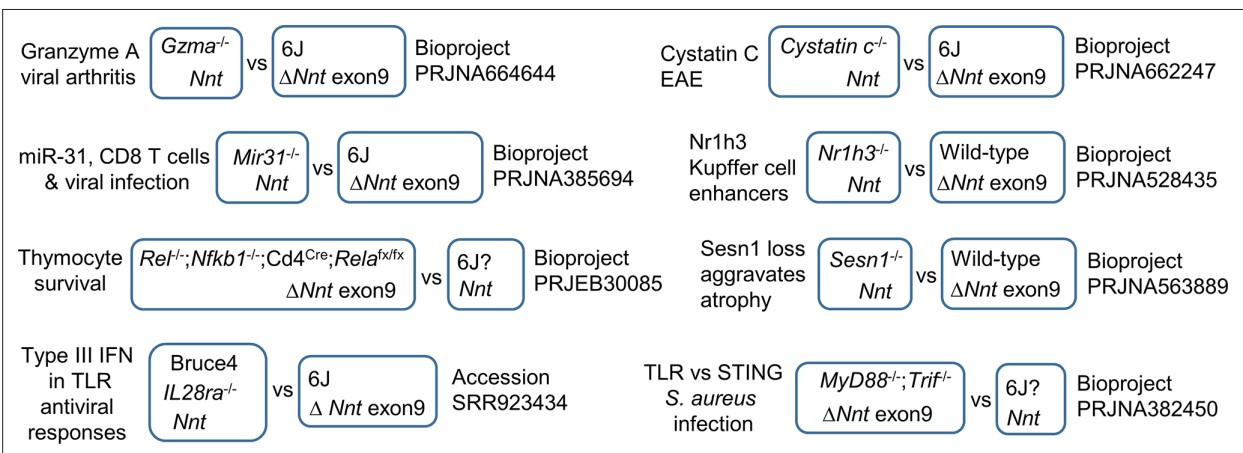

**Figure 8.** k-mer mining of BioProjects where Nnt⁺ mice were compared with *Nnt⁻* mice. The NCBI Sequence Read Archive (SRA) database was interrogated by k-mer mining for BioProjects where (i) some Run Accessions (listing 6J as the mouse strain) had reads compatible with a 6J background (reads for *Nnt* exon 2, but not exon 9) and (ii) other Run Accessions in that BioProject (listing 6J as the mouse strain) had reads not compatible with a 6J background (reads for *Nnt* exons 2 and 9). The methodology is described in *Figure 8—figure supplement 1a*, validated by BLAST alignments (*Figure 8—figure supplement 1b*), with raw data in *Supplementary file 7d and e*.

The online version of this article includes the following figure supplement(s) for figure 8:

**Figure supplement 1.** k-mer mining methodology and validation.

**Figure supplement 2.** Bruce4 ES cell line and *IL28RA⁻/⁻* mouse *Nnt* genotypes.

strain field had RNA-Seq reads that were consistent with 6J (*Nnt-*) (62%), (ii) BioProjects where all the Run Accessions with 'C57BL/6J' in the strain field were not consistent with 6J (*Nnt+*) (23%), and (iii) BioProjects where some Run Accessions with 'C57BL/6J' in the strain field were *Nnt-* and others were *Nnt+* (n = 57; 15%). Thus, 38% (15% plus 23%) of BioProjects had Run Accessions with 'C57BL/6J' strain listings not compatible with a 6J background.

Of the 57 aforementioned BioProjects, 43 had at least one publication associated with the study. These BioProjects were then manually interrogated to identify studies where it was clear (from the paper and the metadata) that comparisons had been made between two groups, where all the Run Accessions in one group were *Nnt+*, and all the Run Accessions in the other group were *Nnt-* (*Figure 8*, *Figure 8—figure supplement 1b*, *Supplementary file 7d and e*). Aside from the CHIKV BioProject described herein, several others emerged (*Figure 8*). For example, *Mir31-/-* mice showed reduced CD8 T cell dysfunction during chronic viral infection when compared to 6J mice (*Moffett et al., 2017*); however, *Mir31-/-* mice were *Nnt+* (*Figure 8*, BioProject PRJNA385694; *Supplementary file 7d*). *Rel-/-;Nfkb1-/-;Cd4Cre;Relafx/fx* mice were compared with 6J mice to implicate RIPK1 and IKK in thymocyte survival (*Webb et al., 2019*); however, the control 6J mice were *Nnt+* (*Figure 8*, BioProject PRJEB30085; *Supplementary file 7d*). Bruce4 ES cells were reported to be on a 6J background (*Ank et al., 2008*) and were derived from a B6 mouse strain congenic for the Thy1.1 allele from an NZB mouse (*Hughes et al., 2007*; *Köntgen et al., 1993*), with these ES cells clearly *Nnt+* (*Figure 8*, accession SRR923434; *Supplementary file 7d*). The reported differences between Bruce4 and 6J genomes were thus likely more to do with the background than with genetic instability (*Hughes et al., 2007*). *Il28ra-/-* mice were generated using Bruce4 cells and were compared with 6J mice (*Ank et al., 2008*); however, RNA-Seq analysis of *Il28ra-/-* mice showed that (like Bruce4 cells) these mice had full-length *Nnt* (*Figure 8—figure supplement 2*). *Myd88-/-;Trif-/-* double knockout mice were compared with 6J mice during infection with *Staphylococcus aureus* (*Scumpia et al., 2017*); however, the 6J (wild-type) mice were *Nnt+* (*Figure 8*, BioProject PRJNA382450; *Supplementary file 7e*). Female *Cystatin C-/-* mice display significantly lower clinical signs of experimental autoimmune encephalomyelitis (EAE) when compared with 6J mice (*Hoghooghi et al., 2020*); however, *Cystatin C-/-* mice were all *Nnt+* (*Figure 8*, BioProject PRJNA662247; *Supplementary file 7d*). Deletion of *Nr1h3* resulted in reduced chromatin access at a large fraction of Kupffer-cell-specific enhancers (*Sakai et al., 2019*); however, *Nr1h3-/-*, but not the wild-type control, were all *Nnt+* (*Figure 8*, BioProject PRJNA528435; *Supplementary file 7d*). *Sesn1-/-* mice were used to show that loss of Sestrin1 aggravates disuse-induced muscle atrophy when compared with 6J mice (*Segalés et al., 2020*); however, *Sesn1-/-* mice were all *Nnt+* (*Figure 8*, BioProject PRJNA563889; *Supplementary file 7d*).

Of the 57 BioProjects containing *Nnt+* and *Nnt+* Run Accessions, several contained comparisons in which one group contained a combination of *Nnt+* and *Nnt-* Run Accessions, while the other group(s) contained either all *Nnt+* or all *Nnt-* Run Accessions (*Supplementary file 7f*).

Whether the *Nnt* differences (or other background gene differences) would have significantly affected the interpretation of phenotypes in the aforementioned studies remains to be established. It is worth noting that herein we have only compared phenotypes of homozygotes (*Figure 7a*), with intermediate phenotypes potentially seen for heterozygotes (*Ronchi et al., 2013*). Our k-mer mining analysis also did not distinguish between *Nnt+/+* and *Nnt+/-*, doing so would require extraction of *Nnt* reads, alignment to the C3H/HeJ genome, and analysis using Sashimi plots (as in *Figure 3d*). Nevertheless, the data argues that differences in *Nnt* or other background gene differences are widely underappreciated in a range of research fields and have the potential to compromise a wide range of studies.

## Discussion

Despite a large body of literature, no clear consensus has emerged regarding the physiological function of GZMA (see 'Introduction'). This lack of consensus might now, at least in part, be explained by the extensive use of *Gzma-/-* mice (*Supplementary file 1*). We show herein that this mouse strain is on a mixed 6J/6N genetic background and contains a full-length *Nnt* gene, with both *Nnt* and other 6N background genes, rather than loss of GZMA expression, responsible for the ameliorated CHIKV arthritis phenotype. Whether all the phenotypes reported for *Gzma-/-* mice (*Supplementary file 1*) are compromised by *Nnt* and/or the mixed background remains unclear and would require new experiments to resolve, similar to those described herein for CHIKV arthritis. However, *Nnt* has been

reported to affect redox regulation and activation in macrophages (*Ripoll et al., 2012*; *Salerno et al., 2019*), with an intact *Nnt* gene conceivably reducing cross-presentation (*Dingjan et al., 2016*; *Nalle et al., 2020*) and CD8 T cell responses (*Oberkampf et al., 2018*). In addition, 6N vs. 6J background differences (as herein) have shown clear phenotypes in a wide range of settings (*Leskov et al., 2017*; *Morales-Hernandez et al., 2018*; *Rao et al., 2020*; *Salerno et al., 2019*; *Vozenilek et al., 2018*; *Williams et al., 2021*; *Wolf et al., 2016*).

The data from *Gzma*<sup>S211A</sup> mice (that were generated using CRISPR on a 6J background) represents the first in vivo assessment of the physiological function of GZMA without the confounding influence of differences in *Nnt* or other genes associated with the mixed genetic background. The results from this analysis support the view that the physiological activity of GZMA is mediated by its protease activity (*Figure 7b*; *Schanoski et al., 2019*). This is an important point because protease-independent functions have been documented for several proteases (*Calhan and Seyrantepe, 2017*; *McNutt et al., 2007*; *Pu et al., 2019*; *Szabo et al., 2016*). One of the proposed activities for GZMA, the binding to TLR9 and potentiation of TLR9 signaling (*Shimizu et al., 2019*), was not reported to require GZMA's protease activity. However, using mice defective in TLR9 signaling, we were unable to find evidence that TLR9 is required for GZMA's pro-inflammatory activity (*Supplementary file 8*). Our studies also support the view that secreted extracellular GZMA has biological activity in the absence of perforin (or GZMB) (*Figure 6—figure supplement 1*). This contrasts with the traditional view of GZMA as a mediator of cell death, which requires perforin to deliver GZMA to the cytoplasm, where a series of target molecules are cleaved (*Liesche et al., 2018*; *Wu et al., 2019*; *Zhou et al., 2020*). Cleavage of SET complex proteins (*Mandrup-Poulsen, 2017*; *Mollah et al., 2017*) would similarly require delivery of GZMA to the cytoplasm. Although we cannot formally exclude translocation of circulating GZMA into the cytoplasm via some unknown mechanism, the *Gzma*<sup>S211A</sup> RNA-Seq experiment does ostensibly exclude perforin as NK cells only produce perforin (and GZMB) protein after ≈24 hr of appropriate stimulation (*Fehniger et al., 2007*). Extracellular target candidates for GZMA include protease-activated receptors (*Hansen et al., 2005*; *Sower et al., 1996*; *Suidan et al., 1994*; *Suidan et al., 1996*) and may also include pro-IL-1 (*Hildebrand et al., 2014*; *Figure 7c*) as pro-IL-1 can become extracellular when cells lyse (*Afonina et al., 2015*). Overall one might speculate that in such settings NK-derived GZMA synergizes with type I IFN responses to act as a systemic alarmin (*Figure 7b*). *Gzma*<sup>S211A</sup> mice will provide an invaluable tool for future studies into further refining our understanding of the role and molecular targets of GZMA.

The ability to reduce CHIKV arthritis in 6J mice with MitoTEMPO might suggest such antioxidant drugs have potential utility as anti-inflammatory treatments for alphaviral arthritides (*Suhrbier et al., 2012*; *Zaid et al., 2021*). However, MitoTEMPO treatment may simply be correcting (at least in part, *Figure 4e*) the *Nnt* defect in 6J mice by scavenging excess mitochondrial ROS arising from the loss of functional NNT (*Ward et al., 2020*). The argument that similar antioxidant treatments would be effective in human diseases may thus not be overly compelling, given that most humans have a functional *Nnt* gene. Perhaps noteworthy is that >300 papers listed in PubMed use MitoTEMPO in 6J mice, with many reporting effective disease amelioration with MitoTEMPO treatments, for example (*Aoyama et al., 2012*; *Li et al., 2018*; *To et al., 2020*; *Vincent et al., 2020*; *Wu et al., 2020*). Unfortunately, antioxidants have not shown clear benefits in human clinical trials (*Casas et al., 2020*; *Jiang et al., 2020*; *Kovacic, 2020*; *Steinhubl, 2008*).

The Jackson Laboratory generated the 6J inbred mouse strain in the 1920s–1930s, with this mouse strain the most frequently used mouse strain in biomedical research. Although differences in the *Nnt* gene (or other background genes) have previously been reported as underappreciated in metabolism research (*Fontaine and Davis, 2016*), the data herein and elsewhere argue that this issue extends to other areas of research (*Bourdi et al., 2011*; *Leskov et al., 2017*; *McCambridge et al., 2019*; *Morales-Hernandez et al., 2018*; *Rao et al., 2020*; *Ripoll et al., 2012*; *Salerno et al., 2019*; *Vozenilek et al., 2018*; *Williams et al., 2021*; *Wolf et al., 2016*), with age effects also reported (*Ghosh et al., 2014*; *Ubaida-Mohien et al., 2019*). Of concern was that ≈27% of SRA Run Accessions and ≈38% of BioProjects listing C57BL/6J as the mouse strain had *Nnt* sequence data not consistent with a pure 6J background. Mouse strain listing errors or inadequate backcrossing to 6J would thus appear to be common for SRA RNA-Seq submissions. The full extent to which *Nnt* and/or genetic backgrounds have complicated interpretation of knockout mouse studies remains to be addressed, but may require extensive new experiments such as those described herein for GZMA.

# Materials and methods

**Key resources table**

| Reagent type (species) or resource | Designation | Source or reference | Identifiers | Additional information |
|---|---|---|---|---|
| Strain, strain background (chikungunya virus) | CHIKV | Dr. P. Roques (CEA, Fontenay-aux-Roses, France) | KT449801.1 | Isolate LR2006-OPY1 |
| Chemical compound, drug | TRIzol | Sigma-Aldrich | Cat# 15596026 | |
| Chemical compound, drug | MitoTEMPO | Sigma-Aldrich | Cat# 1334850-99-5 | |
| Commercial assay, kit | TruSeq RNA Sample Prep Kit (v2) | Illumina | SCR_010233 | |
| Commercial assay, kit | TruSeq Stranded mRNA library preparation kit | Illumina | SCR_010233 | |
| Commercial assay, kit | QIAamp DNA Micro Kit | QIAGEN | Cat# 56304 | |
| Commercial assay, kit | iScript cDNA Synthesis Kit | Bio-Rad | Cat# 1708890 | |
| Commercial assay, kit | Q5 Hot Start High-Fidelity DNA Polymerase | NEB | Cat# M0493S | Enzyme |
| Other | Illumina HiSeq 2000 Sequencer | Illumina | RRID:SCR_010233 | Sequencing platform |
| Other | NextSeq 550 | Illumina | RRID:SCR_016381 | Sequencing platform |
| Other | NovaSeq 6000 | Illumina | RRID:SCR_016387 | Sequencing platform |
| Software, algorithm | k-mer_mining_SRA | GitHub | | https://github.com/CameronBishop/k-mer_mining_SRA |
| Cell line (*Cercopithecus aethiops*) | Vero cells | ATCC | RRID:CVCL_0059 | |
| Cell line (*Aedes albopictus*) | C6/36 cells | ATCC | RRID:CVCL_Z230 | |
| Strain, strain background (*Mus musculus*) | C57BL/6J | Animal Resources Centre (Canning Vale, WA, Australia) | IMSR_JAX:000664 | |
| Strain, strain background (*M. musculus*) | C57BL/6N | The Jackson Laboratory | Stock no. 005304 | |
| Strain, strain background (*M. musculus*) | C57BL/6-*Gzma*[-/-] | Peter MacCallum Cancer Centre, Melbourne, Victoria, Australia | | Knockout mouse |
| Strain, strain background (*M. musculus*) | C57BL/6J-*Gzmb*[-/-] | Peter MacCallum Cancer Centre, Melbourne, Victoria, Australia | | Knockout mouse |
| Strain, strain background (*M. musculus*) | C57BL/6J-*Gzma*[S211A] | The Australian Phenomics Network, Monash University, Melbourne, Australia (this paper) | | Mutant mouse |
| Strain, strain background (*M. musculus*) | C57BL/6N[ΔNnt8-12] | The Australian Phenomics Network, Monash University, Melbourne, Australia (this paper) | | Knockout mouse |

*Continued on next page*

*Continued*

| Reagent type (species) or resource | Designation | Source or reference | Identifiers | Additional information |
|---|---|---|---|---|
| Strain, strain background (*M. musculus*) | C57BL/6J-*Ifnar*$^{-/-}$ | Dr P. Hertzog (Monash University, Melbourne, Australia) | Knockout mouse | |
| Strain, strain background (*M. musculus*) | C57BL/6-*Il28ra*$^{-/-}$ | Bristol-Myers Squibb (PMID:25901316) | Knockout mouse | |
| Sequence-based reagent | Nnt_RTPCR_F1 | This paper | PCR primers | AACAGTGCAAGGAGGTGGAC |
| Sequence-based reagent | Nnt_RTPCR_R1 | This paper | PCR primers | GTGCCAAGGTAAGCCACAAT |
| Software, algorithm | FastQC | Babraham Institute | RRID:SCR_014583 | |
| Software, algorithm | MultiQC | PMID:27312411 | RRID:SCR_014982 | |
| Software, algorithm | Cutadapt | DOI: https://doi.org/10.14806/ej.17.1.200 | RRID:SCR_011841 | |
| Software, algorithm | STAR | PMID:23104886 | RRID:SCR_004463 | |
| Software, algorithm | RSEM | PMID:21816040 | RRID:SCR_013027 | |
| Software, algorithm | EdgeR | PMID:27280887 | RRID:SCR_012802 | |
| Software, algorithm | 'Ingenuity Pathway Analysis' (IPA) | QIAGEN | RRID:SCR_008653 | |
| Software, algorithm | Cytoscape | PMID:14597658 | RRID:SCR_003032 | |
| Software, algorithm | STRING | PMID:30476243 | RRID:SCR_005223 | |
| Software, algorithm | 'Gene Set Enrichment Analysis' (GSEA) | PMID:16199517 | RRID:SCR_003199 | |
| Software, algorithm | 'Integrative Genomics Viewer' (IGV) | PMID:21221095 | RRID:SCR_011793 | |
| Software, algorithm | minimap2 | PMID:29750242 | RRID:SCR_018550 | |
| Software, algorithm | BigQuery | Google Cloud Platform | RRID:SCR_001011 | |
| Software, algorithm | fasterq-dump | SRA tool kit | sra-tools v 2.9.1 | |

## Cell lines and CHIKV

Vero (ATCC#: CCL-81) and C6/36 cells (ATCC# CRL-1660) were cultured as described (*Nguyen et al., 2020*). Cells were checked for mycoplasma using MycoAlert Mycoplasma Detection Kit (Lonza, Basel, Switzerland). FBS was checked for endotoxin contamination using RAW264.7-HIV-LTR-luc cells (*Johnson et al., 2005*) before purchase. CHIKV (isolate LR2006-OPY1; GenBank KT449801.1; DQ443544.2) was a kind gift from Dr. P. Roques (CEA, Fontenay-aux-Roses, France), was propagated in C6/36 cells, and titers determined by CCID$_{50}$ assays (*Nguyen et al., 2020*). Virus was also checked for mycoplasma (*La Linn et al., 1995*).

## Mice and CHIKV infections

C57BL/6J mice were purchased from Animal Resources Centre (Canning Vale, WA, Australia). C57BL/6N mice were purchased from The Jackson Laboratory (stock no. 005304). *Gzma*$^{-/-}$ mice were generated as described (*Ebnet et al., 1995*) and were provided by the Peter MacCallum Cancer Centre, Melbourne, Victoria, Australia. *Gzmb*$^{-/-}$ mice were generated as described (*Müllbacher et al., 1999*) and were backcrossed onto C57BL/6J mice a total of 12 times and were provided by the Peter MacCallum Cancer Centre. The Australian Phenomics Network, Monash University, Melbourne, Australia, used CRISPR to generate (i) *Gzma*$^{S211A}$ mice on a 6J background and (ii) 6N$^{\Delta Nnt8\text{-}12}$ mice on a 6N background. *Ifnar*$^{-/-}$ mice (*Yan et al., 2020*) were kindly provided by Dr. P. Hertzog (Monash

University). *Il28ra*[-/-] mice (*Ank et al., 2008*) were kindly provided by Bristol-Myers Squibb (*Souza-Fonseca-Guimaraes et al., 2015*). All GMO mice were bred in-house at QIMR Berghofer MRI.

Female mice 6–8 weeks old were infected with $10^4$ CCID$_{50}$ CHIKV (isolate LR2006 OPY1) s.c. into each hind foot, with foot measurements and viremia determined as described (*Gardner et al., 2010*; *Nguyen et al., 2020*; *Prow et al., 2018*).

## RNA-Seq of feet of CHIKV-infected GMO mice

Mice were infected with CHIKV, feet collected on day 6, and RNA samples prepared as described previously (*Hazlewood et al., 2021*; *Rawle et al., 2021*; *Wilson et al., 2017*) with minor modifications. Briefly, library preparation and sequencing were conducted by the Australian Genome Research Facility (Melbourne, Australia) (*Hazlewood et al., 2021*) or were conducted in-house (*Rawle et al., 2021*). RNA concentration and quality was measured using TapeStation D1K TapeScreen assay (Agilent). cDNA libraries were prepared using a TruSeq RNA Sample Prep Kit (v2) (Illumina Inc, San Diego, USA), which included isolation of poly-adenylated RNA using oligo-dt beads or total RNA library preparation with rRNA depletion (NEBNext Ultra II Directional RNA Library Prep Kit for Illumina and NEBNext rRNA Depletion Kit v2). Paired end reads were generated using the Illumina HiSeq 2000 Sequencer (Illumina Inc) (100 bp) or Illumina NextSeq 550 platform (75 bp). Per-base sequence quality for >90% bases was above Q30 for all samples. Raw sequencing reads were assessed using FastQC (v0.11.8) (*Simons, 2010*) and MultiQC (v1.7) (*Ewels et al., 2016*) and trimmed using Cutadapt (v2.3) (*Martin, 2011*) to remove adapter sequences and low-quality bases. Trimmed reads were aligned to the GRCm38 primary assembly reference and the GENCODE M23 gene model using STAR aligner (v2.7.1a) (*Dobin et al., 2013*), with more than 95% of reads mapping to protein coding regions. Counts per gene were generated using RSEM (v1.3.1) (*Li and Dewey, 2011*), and differential expression analysis was undertaken using EdgeR in Galaxy (*Varet et al., 2016*) with default settings and a count sum $\geq 1$ filter applied. Counts were normalized using the TMM method and modeled using the likelihood ratio test, glmLRT().

## Bioinformatic analyses

Pathway analysis of DEGs in direct and indirect or direct-only interactions was investigated using IPA (QIAGEN; *Shannon et al., 2003*). Enrichment for biological processes, molecular functions, KEGG pathways, and other gene ontology categories in DEG lists was elucidated using the STRING database (*Szklarczyk et al., 2019*) in Cytoscape (v3.7.2) (*Shannon et al., 2003*). GSEA (*Subramanian et al., 2005*) was performed on a desktop application (GSEA v4.0.3) (http://www.broadinstitute.org/gsea/) to look for enrichment of DEGs in full gene sets preranked by fold change.

## WGS of *Gzma*[-/-] mice

QIAamp DNA Micro Kit (QIAGEN) was used to purify genomic DNA from *Gzma*[-/-] mice spleen as per the manufacturer's instructions. DNA was sent to the Australian Genome Research Facility (AGRF) for WGS using the Illumina NovaSeq platform with 150 bp paired end reads. The primary sequence data was generated using the Illumina bcl2fastq 2.20.0.422 pipeline. Reads were mapped to the mm10 genome assembly (GRCm38) using BWA-mem, and .bam files were provided. Mapped reads were viewed in Integrative Genomics Viewer (IGV) (*Robinson et al., 2011*) and 6N features identified manually based on previous publications (*Mekada et al., 2015*; *Simon, 2013*).

## Alignment to mouse genomes

FastQ files were generated as described or were downloaded from the SRA using Aspera. Reads were trimmed using Cutadapt (*Martin, 2011*) and mapped using STAR aligner (v2.7.1a) for RNA-Seq or minimap 2 (*Li, 2018*) for WGS data. IGV was used to visualize reads mapping to the *Nnt* gene after mapping to the GRCm38 primary assembly reference for the truncated version of the gene and to the mouse C3H_HeJ_v1 reference (GCA_001632575.1) to observe full-length *Nnt*.

## *Nnt* RT-PCR

RT-PCR was undertaken essentially as described using the primer set (F1 AACAGTGCAAGGAGGT-GGAC, R1 GTGCCAAGGTAAGCCACAAT) (Integrated DNA Technologies; *Huang et al., 2006*). RNA was extracted from testes using TRIzol (Sigma-Aldrich) according to the manufacturer's instructions.

cDNA was generated using iScript cDNA Synthesis Kit (Bio-Rad) and Q5 Hot Start High-Fidelity DNA Polymerase (NEB) was used for PCR.

## Histology

Histology and H&E staining were undertaken as described (*Rawle et al., 2021*). Sections were scanned using Aperio AT Turbo (Aperio, Vista, CA) and analyzed using Aperio Image-Scope software (Leica Biosystems, Mt Waverley, Australia) (v10). Quantitation using Positive Pixel Count v9 was used to generate blue/red pixel ratios as a measure of leukocyte infiltrates, as described (*Poo et al., 2014*).

## MitoTEMPO treatment

Mice were injected i.v. daily, on the indicated day post CHIKV infection, with 62.5 µg of MitoTEMPO (Sigma-Aldrich) in 150 µl of PBS.

## RNA-Seq of poly(I:C) injection for *Gzma*^S211A vs. 6J mice

Age-matched female *Gzma*^S211A and 6J mice were injected i.v. with 250 µg of poly(I:C) in 150 µl of PBS. After 6 hr, mice were euthanized, spleen and whole feet were harvested, and RNA isolated as described previously (*Prow et al., 2019*; *Wilson et al., 2017*). Three RNA pools were generated for each mouse strain, whereby each pool contained equal amounts of RNA from four feet from four mice, or two spleens from two mice. RNA-Seq of polyadenylated RNA was then undertaken in-house at QIMR Berghofer MRI. RNA integrity was assessed using the TapeStation system (Agilent Technologies), and libraries were prepared using the TruSeq Stranded mRNA library preparation kit (Illumina). Sequencing was performed on the Illumina NextSeq 550 platform with 75 bp paired end reads. Per-base sequence quality for >92% bases was above Q30 for all samples. Raw sequencing reads were then processed as above.

## k-mer mining

An exact-match (31 mer) k-mer mining approach was used to identify RNA-Seq read files (Accessions) with C57BL/6J mice listed as the mouse strain, but where *Nnt* reads were incompatible with 6J background. Metadata associated with the National Center for Biotechnology Information's SRA was screened using the Google Cloud Platform's BigQuery service with the Structured Query Language (SQL) command: SELECT m.bioproject, m.acc, m.sample_name, m.platform, m.mbytes, m.mbases FROM nih-sra-datastore.sra.metadata as m, UNNEST (m.attributes) as a WHERE m.organism = '*Mus musculus*' and m.assay_type = 'RNA-Seq' and a.v = 'C57BL/6J.' Technical replicates for the same sample were collapsed by taking only the first accession for each Biosample. Accessions were then filtered on the basis of their compressed size so that only those between 200 Mb and 1500 Mb were retained; we found that read files of this size provided adequate sequencing depth to detect *Nnt* exon reads. Accessions were sorted according to BioProject and used as input for a bioinformatics pipeline executed on the Google Cloud Platform, which allowed access to the 'SRA in the cloud' database. A copy of our Bash script to automate the pipeline is available at https://github.com/CameronBishop/k-mer_mining_SRA (*Bishop, 2021*). Accession read files were converted to FastQ format using fasterq-dump (SRA tool kit). BBduk version 38.87 (*Bushnell, 2020*) was used with default parameters to screen each read for sequence homology to exons 2 and 9 of the *Nnt* gene. Reads sharing at least one 31-mer with either exon were counted as a 'match' for that exon. FastQ files with at least 10 matches to exon 2 and 0 matches to exon 9 were classed as consistent with a 6J genotype (truncated *Nnt*), while FastQ files with at least 10 matches to each exon were classed as not consistent with a 6J genotype. Results were curated using BigQuery to confirm that for each Accession's metadata entry the 'strain_sam' field (or equivalent) of the metadata 'Attributes' table was listed as C57BL/6J.

BioProjects were identified where some read files contained exon 2 reads and no exon 9 reads, whereas others contained both exon 2 and exon 9 reads. The literature and Gene Expression Omnibus submissions associated with these BioProjects were then consulted to identify BioProjects where mice with full-length *Nnt* had been compared with mice with truncated *Nnt*.

### Determination of GZMA levels in mouse serum samples

Mouse serum was collected in Microvette 500 Z gel tubes (Sarstedt) with GZMA levels determined using a GZMA ELISA kit (MyBioSource, San Diego, CA, MBS704766) according to the manufacturer's instructions.

### Statistics

Statistical analysis of experimental data was performed using IBM SPSS Statistics for Windows, version 19.0. The $t$-test was performed when the difference in variances was <4, skewness was >-2, and kurtosis was <2, otherwise the Kolmogorov–Smirnov test was used.

### Data, code, and GMO mouse availability

All data are provided in the article and accompanying supplementary files. Raw sequencing data generated for this publication has been deposited in the NCBI SRA. RNA-Seq NCBI BioProjects: (i) 6J + CHIKV vs. 6J mock infection, day 2 and day 7 feet (PRJNA431476); $Gzma^{-/-}$ + CHIKV vs. 6J + CHIKV, day 6 feet (PRJNA664644); (ii) $6N^{\Delta Nnt8-12}$ + CHIKV vs. 6N + CHIKV, day 6 feet (PRJNA779556); (iii) 6J + MitoTEMPO + CHIKV vs. 6J + PBS + CHIKV (PRJNA779556); and (iv) 6J $Gzma^{S211A}$ + poly(I:C) vs. 6J + poly(I:C) feet and spleen 6 hr (PRJNA666748). WGS of $Gzma^{-/-}$ mice NCBI BioProject PRJNA664888.

A copy of our code to automate the k-mer mining pipeline is available at https://github.com/CameronBishop/k-mer_mining_SRA (*Bishop, 2021*; copy archived at swh:1:rev:372d29b02972d96e8eff7b-6c431883ea8dfb5c50). CRISPR GMO mouse lines $6N^{\Delta Nnt8-12}$ and $Gzma^{S211A}$ are available on request.

## Acknowledgements

We thank the following staff at QIMR Berghofer MRI for their assistance; animal house staff, Dr I Anraku (BSL3 facility manager), Dr R Johnston (Bioinformatics), and Dr Viviana Lutzky (for proof reading). We thank Dr Dion Kaiserman (Monash University, Australia) for supplying recombinant GZMA. We also thank Dr Mark Heise (University of North Carolina) for valuable discussions.

## Additional information

#### Funding

| Funder | Grant reference number | Author |
|---|---|---|
| National Health and Medical Research Council | APP1141421 | Andreas Suhrbier |
| National Health and Medical Research Council | APP1173880 | Andreas Suhrbier |

The funders had no role in study design, data collection and interpretation, or the decision to submit the work for publication.

#### Author contributions

Daniel J Rawle, Data curation, Formal analysis, Investigation, Methodology, Project administration, Supervision, Visualization, Writing – review and editing; Thuy T Le, Kexin Yan, Investigation; Troy Dumenil, Cameron Bishop, Data curation, Formal analysis, Visualization, Writing – review and editing; Eri Nakayama, Investigation, Conceptualization; Phillip I Bird, Conceptualization, Funding acquisition, Methodology, Resources, Writing – review and editing; Andreas Suhrbier, Conceptualization, Data curation, Formal analysis, Funding acquisition, Methodology, Project administration, Supervision, Visualization, Writing – original draft, Writing – review and editing

#### Author ORCIDs

Troy Dumenil http://orcid.org/0000-0002-3032-8360
Cameron Bishop http://orcid.org/0000-0002-5710-9942
Andreas Suhrbier http://orcid.org/0000-0001-8986-9025

### Ethics

All mouse work was conducted in accordance with the "Australian code for the care and use of animals for scientific purposes" as defined by the National Health and Medical Research Council of Australia. Mouse work was approved by the QIMR Berghofer Medical Research Institute animal ethics committee (P2235 A1606-618M), with infectious CHIKV work conducted in a biosafety level-3 (PC3) facility at the QIMR Berghofer MRI (Australian Department of Agriculture, Water and the Environment certification Q2326 and Office of the Gene Technology Regulator certification 3445).

### Decision letter and Author response

Decision letter https://doi.org/10.7554/eLife.70207.sa1
Author response https://doi.org/10.7554/eLife.70207.sa2

## Additional files

### Supplementary files

• Supplementary file 1. Summary of studies using *Gzma*[-/-] mice. Compilation of studies employing *Gzma*[-/-] or *Gzma*[-/-] *Gzmb*[-/-] double KO mice that reveal a phenotype or do not show a phenotype.

• Supplementary file 2. RNA-Seq of *Gzma*[-/-]+ chikungunya virus (CHIKV) vs. 6J + CHIKV and 6J + CHIKV vs. 6J mock infection. Datasets for RNA-Seq and Ingenuity Pathway Analysis (IPA) of mice feet day 6 post infection from CHIKV-infected *Gzma*[-/-] mice vs. CHIKV-infected 6J mice and of mice feet day 7 post infection from 6J + CHIKV vs. 6J mock infection.

• Supplementary file 3. Genetic differences between *Gzma*[-/-] and C57BL/6J. Genetic differences potentially involved in inflammation or arthritis are indicated. *Nnt* not included. Differences in introns not included.

• Supplementary file 4. RNA-Seq for 6N*Nnt*Δ8-12+ chikungunya virus (CHIKV) vs. 6N + CHIKV. Datasets for RNA-Seq and Ingenuity Pathway Analysis (IPA) of mice feet day 6 post infection; 6N*Nnt*Δ8-12+ CHIKV vs. 6N + CHIKV.

• Supplementary file 5. RNA-Seq of chikungunya virus (CHIKV)-infected mice treated with MitoTEMPO. Datasets for RNA-Seq and Ingenuity Pathway Analysis (IPA) of mice feet day 6 post infection; 6J + MitoTEMPO + CHIKV vs. 6J + PBS + CHIKV.

• Supplementary file 6. RNA-Seq of *Gzma*[S211A] vs. 6J mice injected with polyinosinic:polycytidylic acid (poly(I:C)). Datasets for RNA-Seq, Ingenuity Pathway Analysis (IPA), and Cytoscape analyses of mice feet and spleen taken 6 hr after injection with poly(I:C); *Gzma*[S211A] vs. 6J.

• Supplementary file 7. k-mer mining of the NCBI Sequence Read Archive (SRA). Datasets for exact-match (31 mer) k-mer mining approach to identify full-length Nnt reads (6N background) in accessions listing C57BL/6J mice as the mouse strain.

• Supplementary file 8. No evidence for TLR9 involvement. (a) *Tlr9*[-/-] and 6J mice were injected intraplantar into the feet with 5 μg recombinant mouse granzyme A (GZMA) in 20 μl and foot swelling measured over time as described (*Schanoski et al., 2019*) (n = 4 mice and four feet per group; statistics by Kolmogorov–Smirnov tests). *Tlr9*[-/-] mice were derived from 129/Ola × C57BL/6F1 progeny (http://www.myv.ne.jp/obs/index.files/tlr_eng.htm). (b) As for (a) using *Tlr9*[M7Btlr]/Mmjax and 6J mice. *Tlr9*[M7Btlr]/Mmjax mice have a *Tlr9* missense point mutation and do not respond to oligonucleotides containing CpG motifs (https://www.jax.org/strain/014534). (c) *Tlr9*[-/-] mice (like 6J) do not encode the full *Nnt* gene. (d) Female 8–10-week-old *Tlr9*[-/-] and 6J mice (n = 6 mice and 12 feet per group) were infected with chikungunya virus (CHIKV) and feet measured over time. Statistics by Kolmogorov–Smirnov tests. (e) Female C57BL/6J-*Tlr9*[M7Btlr]/Mmjax mice (n = 6 mice and 12 feet per group) were infected as for (d). (f) Viremia for the mice in (d). After GZMA injection, *Tlr9*[-/-] mice showed increased foot swelling (a), whereas C57BL/6J-*Tlr9*[M7Btlr]/Mmjax mice showed no significant difference (b). *Tlr9*[-/-] mice also have the *Nnt* deletion (c); however, they are on a mixed 129/Ola and C57BL/6 background (Hemmi et al. 2000), with 129/SvJ mice showing increased inflammatory infiltrates in certain settings (Hoover-Plow et al., 2008). After CHIKV infection, foot swelling was again increased in *Tlr9*[-/-] (d), but not *Tlr9*[M7Btlr]/Mmjax mice (e). *Tlr9*[-/-] mice did not show an increased viremia (f). These data do not support a contention that TLR9 is required for GZMA's bioactivity.

• Transparent reporting form

• Source data 1. Source data for DNA gel images. Source data for DNA gel images in *Figure 3e*, *Figure 1—figure supplement 1b,c*, and *Figure 4—figure supplement 1e*.

## Data availability

Five supplementary files have been provided which constitute source data for all the results cited in the manuscript. Raw sequencing data was uploaded to SRA: BioProject accessions: PRJNA666748, PRJNA664888, PRJNA664644, PRJNA779556.

The following datasets were generated:

| Author(s) | Year | Dataset title | Dataset URL | Database and Identifier |
|---|---|---|---|---|
| Suhrbier A | 2020 | RNA-Seq of Granzyme A proteolytic site mutant mice injected with Poly (I:C) | https://www.ncbi.nlm.nih.gov/bioproject/PRJNA666748 | NCBI BioProject, PRJNA666748 |
| Suhrbier A | 2020 | Whole genome sequencing of a Granzyme A knock out mouse | https://www.ncbi.nlm.nih.gov/bioproject/PRJNA664888 | NCBI BioProject, PRJNA664888 |
| Suhrbier A | 2020 | RNA-Seq of Granzyme A knockout mice infected with chikungunya Virus | https://www.ncbi.nlm.nih.gov/bioproject/PRJNA664644 | NCBI BioProject, PRJNA664644 |
| Suhrbier A | 2021 | RNA_Seq of C57BL/6N Nnt knockout mice and MitoTEMPO treatment of C57BL/6J mice infected with CHIKV | https://www.ncbi.nlm.nih.gov/bioproject/PRJNA779556 | NCBI BioProject, PRJNA779556 |

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
