## [Editor Report]

This paper is of great interest as a serendipitous discovery that, in the course of investigating the physiological role of granzyme A, has revealed the significance of the *Nnt* gene mutation in the inflammatory responses in mouse models. For many researchers in the fields of medicine and biology using C57BL/6 mice, the data obtained in this study will provide a useful opportunity to revisit previous findings and to gain new insights.

---

## [Decision Letter]

**Decision letter after peer review:**

Thank you for submitting your article "Widespread discrepancy in Nnt genotypes complicate granzyme A and other knockout mouse studies" for consideration by *eLife*. Your article has been reviewed by 2 peer reviewers, and the evaluation has been overseen by a Reviewing Editor and Carlos Isales as the Senior Editor. The reviewers have opted to remain anonymous.

Essential revisions:

1) Please provide histological data of the feet of B6N, B6N-NntΔexon8-12, and B6J mice administered MitoTEMPO after infection with CHIKV, to demonstrate the difference in leukocyte infiltration. (as suggested by Reviewer #2)

2) The ROS levels in T cells should be examined. Is there difference between mice with or lacking NNT activity? (as suggested by Reviewer #2)

3) Please provide some data and/or discussion about the possibility that the truncated Nnt allele in heterozygosis may generate an intermediate phenotype. (as suggested by Reviewer #1)

4) Please revise and improve the manuscript according to the comments by both reviewers.

*Reviewer #1 (Recommendations for the authors):*

– Although the text is very explanative, the excessive use of abbreviations (together with the absence of an abbreviation list since it is not required by the journal) make it sometimes difficult to read fluently. I suggest reducing the use of abbreviations whenever possible.

– Nicotinamide nucleotide transhydrogenase is the name of the gene encoding the protein. Indeed, "NNT" has also been extensively used to refer to the protein. However, in enzyme databases the official name of this enzyme is proton-translocating NAD(P)+ transhydrogenase (EC 7.1.1.1). I suggest adding this information to the introduction.

– The Methods section does not seem logically organized. Please, re-order the topics, for example by the type of procedure (e.g. putting any treatment just after the description of mice, and grouping all procedures involving RNA/rtPCR, and then genome and bioinformatics).

– The presence of the truncated Nnt allele in heterozygosis may generate an intermediate phenotype (as suggested in PMID 27474736). Did the assessment carried out on SRA-NCBI data or even the GzmA-/- sequencing allow to detect heterozygous carriers? The authors may wish to include some data and/or discussion on this issue.

*Reviewer #2 (Recommendations for the authors):*

Even though the conclusions of this paper are well supported by ample data, it would be more convincing if the authors could provide the following data.

Histological section of the feet of B6N, B6N-NntΔexon8-12, and B6J mice administered MitoTEMPO after infection with CHIKV to demonstrate the difference in leukocyte infiltration.

The difference in ROS levels in T cells of mouse strains due to presence or absence of NNT activity.

Validation of RNA-seq data by quantitative RT-PCR.

The authors should address whether or not is C57BL/6N-GzmaS211A knock-in mice is available to other researchers.

---

## [Author Response]

Essential revisions:1) Please provide histological data of the feet of B6N, B6N-NntΔexon8-12, and B6J mice administered MitoTEMPO after infection with CHIKV, to demonstrate the difference in leukocyte infiltration. (as suggested by Reviewer #2)

This data has been provided in new Figure 4c,d, 5c,d,e and f, and Supplementary Figures 6 and 7. Actually the results indicated a somewhat more nuanced outcome with 6N^NntΔexon8-12^ mice showing increased edema rather than infiltrates, and MitoTEMPO treatment showing reduced edema and infiltrates. To unravel this issue we undertook RNA-Seq analyses for the two comparisons to produce a new Figure 4e. The results highlight that *Nnt* and the 6N background both likely have a role in limiting arthritic inflammation. Adjustments to reflect this new data have been made throughout the manuscript including the title.

2) The ROS levels in T cells should be examined. Is there difference between mice with or lacking NNT activity? (as suggested by Reviewer #2)

Given that 6N^NntΔexon8-12^ mice did not show differences in cellular infiltrates, it is clear our speculation on NNT regulating T cell migration in this setting was misplaced. That ROS regulates T cell migration is established, and we show that MitoTEMO also inhibits cellular infiltration; however, in this model at least, NNT emerges only to significantly ameliorate arthritic edema. We have thus removed our speculation on T cell migration and NNT, and have replaced this with new histology and RNA-Seq data, which details the cytokines and chemokines affected by NNT and MitoTEMPO (see Figure 4e). We have also been unable to see differences in ROS levels using a number of florescent dyes in 6N^NntΔexon8-12^ vs 6N splenic T cells under a range of in vitro conditions; levels are actually quite low as activation causes a metabolic shift to glycolysis. NNT has been reported independently by two groups to regulate mitochondrial ROS in macrophages, with APC function thus a more likely target for NNT.

3) Please provide some data and/or discussion about the possibility that the truncated Nnt allele in heterozygosis may generate an intermediate phenotype. (as suggested by Reviewer #1)

This is a good point and we have attempted to compare the foot swelling phenotype for 6N^NntΔexon8-12^ homozygotes and heterozygotes. However, given the inherent variability in foot swelling, this is not a good model to obtain clear answers on intermediate phenotypes. We are currently exploring this issue in another model, specifically SARS-CoV-2 lethality in K18-hACE2 mice, where the phonotype is considerably less variable.

The issue is clearly also very pertinent to interpreting the implications of k-mer mining results, where we did not distinguish between homozygotes and heterozygotes. At the end of the k-mer mining section we have thus added:

“It is worth noting that herein we have only compared phenotypes of homozygotes (Figure 7a), with intermediate phenotypes potentially seen for heterozygotes (Ronchi et al., 2013). Our k-mer mining analysis also did not distinguish between *Nnt^+/+^* and *Nnt^+/-^*genotypes, doing so would require extraction of *Nnt* reads, alignment to the C3H/HeJ genome, and analysis using Sashimi plots (as in Figure 3d)”.

Reviewer #1 (Recommendations for the authors):– Although the text is very explanative, the excessive use of abbreviations (together with the absence of an abbreviation list since it is not required by the journal) make it sometimes difficult to read fluently.

We have added an abbreviation list that the journal can hopefully add as a footnote to address this issue. Unfortunately, this is a complex story with many players, with many abbreviations difficult to avoid.

– Nicotinamide nucleotide transhydrogenase is the name of the gene encoding the protein. Indeed, "NNT" has also been extensively used to refer to the protein. However, in enzyme databases the official name of this enzyme is proton-translocating NAD(P)+ transhydrogenase (EC 7.1.1.1). I suggest adding this information to the introduction.

Good point – added as requested.

– The Methods section does not seem logically organized.

We have reordered the Methods in order of the appearance of data in the figures.

– The presence of the truncated Nnt allele in heterozygosis may generate an intermediate phenotype (as suggested in PMID 27474736). Did the assessment carried out on SRA-NCBI data or even the GzmA-/- sequencing allow to detect heterozygous carriers?

We have discussed this issue and explained how heterozygotes might be identified using RNA-Seq data; however, as a screening approach this would require some extraordinarily large amount of computing power given the size of the SRA data base. We have now clarified that our K-mer mining approach does not provide insights into heter/homozygosity. See response to editor’s comments above.

Reviewer #2 (Recommendations for the authors):Even though the conclusions of this paper are well supported by ample data, it would be more convincing if the authors could provide the following data.Histological section of the feet of B6N, B6N-NntΔexon8-12, and B6J mice administered MitoTEMPO after infection with CHIKV to demonstrate the difference in leukocyte infiltration.

This data has been provided in new Figure 4c,d, 5c,d and Supplementary Figures 6 and 7. Please see responses to editor’s comments above.

The difference in ROS levels in T cells of mouse strains due to presence or absence of NNT activity.

Please see responses to editor’s comments above.

Validation of RNA-seq data by quantitative RT-PCR.

RNA-Seq is now a well characterized methodology and does not suffer from the limited dynamic range of microarray technology, where RT-PCR validation was frequently required. We should also note that we are not making claims regarding expression of any specific individual genes, but are using pathway analyses where the changes of a large number of mRNAs are combined. The only gene were we do make claims is *Nnt* and for this gene we have provided RT PCR data (Figure 3e). For GzmA we also provide protein expression data (Supplementary Figure 1e).

To perhaps address concerns to some extent, we have added our quality control approaches for the RNA-Seq in the Methods, which encompasses both RNA quality, as well as bioinformatic approaches including FastQC, RNAseqQC, with per base sequence quality for >90% bases >Q30 for all samples.

The authors should address whether or not is C57BL/6N-GzmaS211A knock-in mice is available to other researchers.

These GMO lines are available on request – we have added this to the Data code and mouse availability section.